# LT-SNN: Self-Adaptive Spiking Neural Network for Event-based Classification and Object Detection

## Abstract

Spiking neural networks (SNNs) have received increasing attention due to its high biological plausibility and energy efficiency. The binary spike-based information propagation enables efficient sparse computation with event-based computer vision applications. Prior works investigated direct SNN training algorithm to overcome the non-differentiability of spike generation. However, most of the existing works employ a fixed threshold value for the membrane potential throughout the entire training process, which limits the dynamics of SNNs towards further optimizing the performance. The adaptiveness in the membrane potential threshold and the mismatched mechanism between SNN and biological nervous system remain under-explored in prior works. In this work, we propose LT-SNN, a novel SNN training algorithm with self-adaptive learnable potential threshold to improve SNN performance. LT-SNN optimizes the layer-wise threshold value throughout SNN training, imitating the self-adaptiveness of the biological nervous system. To stabilize the SNN training even further, we propose separate surrogate gradient path (SGP), a simple-yet-effective method that enables the smooth learning process of SNN training. We validate the proposed LT-SNN algorithm on multiple event-based datasets, including both image classification and object detection tasks. Equipped with high adaptiveness that fully captures the dynamics of SNNs, LT-SNN achieves state-of-the-art performance with compact models. The proposed LT-SNN based classification network surpasses SoTA methods where we achieved 2.71% higher accuracy together with $10.48\times$ smaller model size. Additionally, our LT-SNN-YOLOv2 object detection model demonstrates 0.11 mAP improvement compared to the SoTA SNN-based object detection.

## 1 Introduction

In the biological nervous system, cortical neurons process information by encoding spatial-temporal inputs into action potentials for spike generation. Inspired by that, spiking neural networks (SNNs) accumulate the membrane potential by extracting information from the input features at each time step, and the resultant binary spikes (0 and 1) provides a sparse and succinct information representation. Such spatial-temporal computation promotes SNN as an attractive AI solution with both biological plausibility and energy efficiency in comparison to the conventional artificial neural networks (ANNs) (He et al., 2016). Furthermore, layer-wise processing with binary spikes elevates the computation efficiency, which benefits the energy-constrained applications such as edge computing.

Under the context of energy-efficient AI applications, the event-based camera or dynamic vision sensors (DVS) have emerged as an attractive and feasible solution for computer vision applications. Compared to the conventional frame-based camera, event cameras independently capture the absolute illumination changes of pixels, resulting in the asynchronous binary stream of events (Gallego et al., 2020). The captured event is characterized by binary pixels and temporal resolutions, leading to highly sparse and energy-efficient visual representations. Such binarized spatial-temporal information naturally fits the computation mechanism of SNNs, bridging the gap between computer vision and neuromorphic computing.

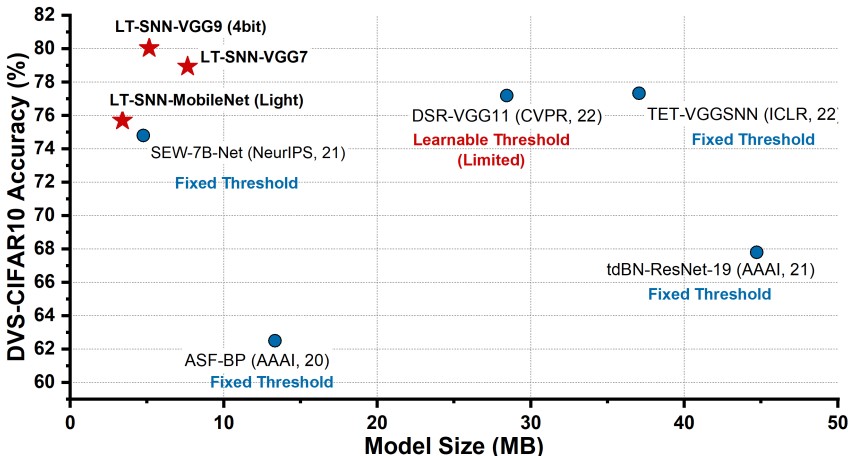

Figure 1: DVS-CIFAR10 classification accuracy of different SNN training methods. The proposed LT-SNN training algorithm achieves the state-of-art accuracy with the compact VGG models.

One of the major bottlenecks of SNN training is the non-differentiability of the spike generation. The infinite gradient of the step function impedes the gradient propagation during backward pass in training. Early research works relied on the ANN-to-SNN conversion (Diehl et al., 2015; Han et al., 2020) to obtain trained SNN models. However, additional training is required and sufficiently high accuracy was not achievable. To that end, various direct training methods have been proposed to obtain a SNN model with one-time training. Empowered by various surrogate gradient (SG) functions (Lee et al., 2016; Wu et al., 2019; Deng et al., 2021), the inaccessible gradient of the spike function is approximated and propagated during learning. However, the inaccurate approximation and heuristic SG selection hurts the training stability with deep models (e.g., ResNet (He et al., 2016)), which further motivated the temporal normalization method (Zheng et al., 2021) and output regularization techniques (Guo et al., 2022; Deng et al., 2021) to smooth the loss.

As the major inspiration of deep learning, the intricate nervous system achieves remarkable performance with a high degree of dynamics. Previous neuroscience works observed the location-dependent potential threshold (Kole & Stuart, 2008) in nervous systems, implying the adaptive firing procedure within the mechanism of spike generation. Inspired by this, some recent works on SNN training introduced the learning dynamics into the training process, albeit to a limited degree. (Fang et al., 2021b) optimized the membrane time constant throughout training, with the requirements of large-sized models. DSR (Meng et al., 2022) proposed the threshold-associated spikes with learnable potential threshold. However, the deterministic ratio between firing range and potential threhsold of DSR limits the adaptiveness of SNN learning. (Sun et al., 2022) removes such constraints by directly passing the gradient of potential threshold through the SG function. Nevertheless, the instability of the straight-through surrogate gradient still results in sub-optimal performance compared to state-of-the-art (SoTA) SNN training with fixed threshold (Deng et al., 2021). Although (Deng et al., 2021) achieved the SoTA performance among prior works, the fixed threshold made the membrane potential to often overshoot, limiting the dynamics of SNNs. The limitations of all prior works motivate us to investigate the following question: *How can we optimize the potential threshold of SNNs with high stability and superior accuracy?*

To answer this question, we propose LT-SNN, a novel self-adaptive **SNN** training algorithm with **L**earnable **T**hreshold. Starting the training from scratch, LT-SNN fully optimizes the potential threshold without introducing any additional scaling or firing constraints. To achieve highly-stable training, we propose a simple-yet-effective technique, namely Separate Gradient Path (SGP). Compared to prior works, the proposed LT-SNN algorithm fully unleashes the advantage of layer-wise adaptive potential threshold, leading to superior performance compared to all prior SNN algorithms. We validate LT-SNN on multiple event-based computer vision datasets with various model architectures. LT-SNN achieves the new state-of-the-art performance with light-weight or quantized models, as shown in Figure 1.

## 2 RELATED WORK

**ANN-to-SNN conversion**   Given the non-differentiability of SNN training, early research works converted high-performance non-spiking ANN model into a spiking version (Diehl et al., 2015; Rueckauer et al., 2016). The major drawbacks of the conversion-based method is high computation latency (high number of time-steps) and the additional efforts for the overall training. Several methods have been proposed to improve the latency of the converted model (Han et al., 2020), but still did not reach the high accuracy of direct SNN training methods.

**Direct SNN training**   The expensive training effort and high latency of the conversion-based schemes promotes the direct SNN training to be an attractive solution. BNTT (Neftci et al., 2019) treats the spatial-temporal computation of SNN as a special version of a recurrent neural network (RNN). The gradient of the spiking process is estimated by surrogate gradient (SG) functions in backpropagation. Driven by accuracy and different model architectures, various SG functions have been proposed, including but not limited to rectangle function (Wu et al., 2019), arctangent (Fang et al., 2021b;a), or triangle functions (Deng et al., 2021). However, the difference between the approximated gradient and the exact gradient largely limits the training stability of SNN, especially for the large-sized models. Motivated by that, tdBN (Zheng et al., 2021) introduces the batch-temporal normalization for deep SNN training, and SEW-ResNet (Fang et al., 2021a) directly passes the gradient via designed residual architecture. In addition to the architecture design, various regularization techniques were presented to stabilize SNN training by rectification of the membrane potential distribution (Guo et al., 2022) and backpropagation with spatio-temporal adjustment (Shen et al., 2022).

**Biological dynamics in SNN**   The biology-inspired SNN training methods are also highlighted in recent research works. (Fang et al., 2021b) introduces the learnable time constant for direct SNN training, but the employed large-sized SNN model and extensive training efforts are computationally expensive. Recently proposed DSR (Meng et al., 2022) optimizes the potential threshold during training by multiplying the binary output spike with the threshold value, where the relationship between the firing range and threshold value is constrained by a deterministic ratio. However, optimizing the potential threshold value with such additional constraints limits the learnability of the SNN model. Furthermore, the threshold-dependent binary spikes require layer-wise scaling operation, which deteriorates the hardware compatibility with expansive high-precision multiplication.

## 3 BASICS OF SPIKING NEURAL NETWORKS

Inspired by the biological nervous system, spiking neural networks process the spatial-temporal information throughout layers. Mathematically, the widely-adopted Leaky Integrate-and-Fire (LIF) model characterizes the membrane potential accumulation with the following ordinary differential equation (ODE):

$$\tau \frac{du(t)}{dt} = -(u(t) - u_{reset}) + I(t),\tag{1}$$

where $u(t)$ represents the membrane potential at time $t$, $V_{reset}$ is the reset potential after the spike operation, and $I_t$ is the synapse current at time t and $\tau$ is the time constant. The ODE in Eq. 1 can be resolved as:

$$u_{t+1} = (1 - \frac{dt}{\tau})u_{t-1} + \frac{dt}{\tau}I,\tag{2}$$

where $I$ is the pre-synaptic "current", characterized by the layer-wise output features of SNN model. Given the iteratively accumulated membrane potential, the binary spikes are generated by the step function:

$$S_t = \theta(u_t - V_{th}) = \begin{cases} 1 & \text{if } u_t \geq V_{th} \\ 0 & \text{otherwise} \end{cases}\tag{3}$$

Where $\theta$ represents the Heaviside step function, and $V_{th}$ represents the membrane potential threshold for spiking neurons. In the forward pass of SNN, the binary post-synaptic spikes are generated when the membrane potential exceeds the potential threshold $V_{th}$.

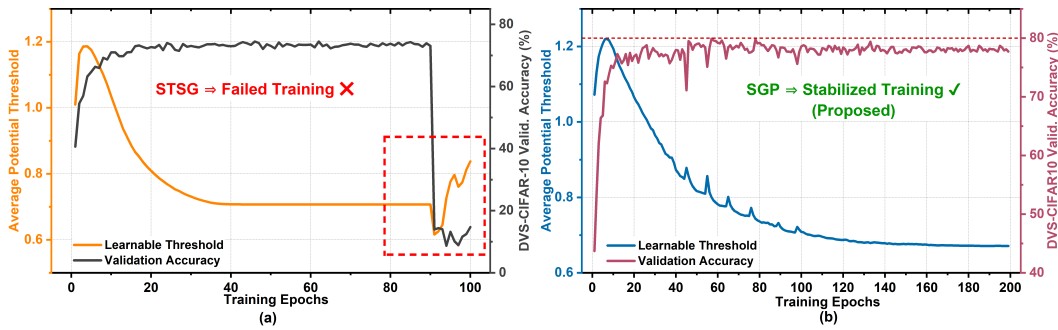

Figure 2: Validation accuracy and potential threshold of direct training SNN on DVS-CIFAR10 dataset with (a) STSG and (b) the proposed separate gradient path (SGP) for LT-SNN.

As demonstrated by spatial-temporal backpropagation (STBP) (Wu et al., 2019), the backward propagation of SNN is characterized as:

$$\frac{\partial L}{\partial W} = \frac{\partial L}{\partial S_t} \frac{\partial S_t}{\partial u_t} \frac{\partial u_t}{\partial I_t} \frac{\partial I_t}{\partial W} \tag{4}$$

As mentioned earlier, the non-differentiable spike function $\theta$ impedes the gradient computation of $\partial S_t / \partial u_t$. The directly-trained SNN incorporates the surrogate gradient function to approximate the intangible Dirac function. In this work, we choose triangle function for gradient approximation:

$$\frac{\partial S_t}{\partial u_t} = \theta'(u_t - V_{th}) = \max(0, 1 - |u_t - V_{th}|) \tag{5}$$

## 4 CHALLENGES OF LEARNING THE THRESHOLD OF MEMBRANE POTENTIAL

As described in Eq. 5, the non-differentiability between the output spike $S$ and the membrane potential $u$ can be alleviated by the surrogate gradient approximation. Mathematically, the gradient of the potential threshold can be approximated with a similar gradient surrogation:

$$\frac{\partial L}{\partial V_{th}} = -\frac{\partial L}{\partial S_t} \frac{\partial S_t}{\partial V_{th}} = -\frac{\partial L}{\partial S_t} \theta'(u_t - V_{th}) \tag{6}$$

The straight-through approximation in Eq. 6 is built upon the following hypothesis:

**Hypothesis 1**: *The surrogate gradient of $\frac{\partial S_t}{\partial u_t}$ and $\frac{\partial S_t}{\partial V_{th}}$ is transferrable, and the identical gradient surrogation is suitable for separate loss landscape with respect to $u_t$ and $V_{th}$.*

In this work, we falsify **Hypothesis 1** by empirically demonstrating the learning instability caused by the straight-through surrogate gradient (STSG). Figure 2(a) depicts the validation accuracy and the layer-wise averaged potential threshold of the VGG-9 model which is trained from scratch with STSG. The drastic change in potential threshold and exploding validation accuracy represents the incompatibility of the STSG learning scheme. In summary, we have the following observation:

**Observation 1:** *The gradient approximation with respect to membrane potential and potential threshold is not interchangeable.*

Different from the STSG scheme, the recent DSR (Meng et al., 2022) scheme computes the gradient of potential threshold with the following threshold-associate firing procedure:

$$S_t = V_{th} \times \theta(u_t - \alpha V_{th}) \tag{7}$$

Where the fixed parameter $\alpha \in [0, 1]$ controls the threshold with respect to the firing range $[0, V_{th}]$. In other words, regardless of the threshold value, the membrane potential has to be a deterministic portion of the total firing range. However, such constraints in the training process largely limits the learnability of SNN. Based on the open-sourced implementation of DSR (Meng et al., 2022), we unleash the optimization constraints of the potential threshold by setting $\alpha = 1.0$. Please note that, keeping the threshold and spike as identical values is commonly adopted in prior works (Deng et al., 2021). As shown in Table 1, the freely-learned potential threshold ($\alpha = 1.0$) exhibits large

Table 1: The performance of DSR (Meng et al., 2022) is largely impacted by the value of $\alpha$. The proposed LT-SNN outperforms DSR with 2.45% higher accuracy on DVS-CIFAR10 dataset.

| Method | Model | Accuracy (%) | $\alpha$ | True Binary Spike |
|---|---|---|---|---|
| DSR (Meng et al., 2022) | VGG-11 | 77.19 | 0.3 | ✗ |
| DSR (Meng et al., 2022) | VGG-11 | 75.29 | 1.0 | ✗ |
| **This work** | VGG-9 | **80.04** | - | ✓ |

accuracy degradation compared to the constrained leanring ($\alpha = 0.3$), which implies the sub-optimal performance of the DSR (Meng et al., 2022) when the potential threshold is unconstrained.

On the other hand, associating the threshold value with the neuron output disables the true binary spikes (0 and 1) of SNN. Specifically, the output generated by Eq. 7 becomes $(0, V_{th})$ instead of a pure binary value $(0, 1)$. In the context of hardware computation, layer-wise varied spikes are equivalent to the layer-wise scaling on top of the true binary output, which often requires high-precision multipliers (Jacob et al., 2018) to maintain the accuracy. Unlike the conventional ANN with one-time scaling operation, the temporal computation of SNN requires step-wise high-precision multiplication, which could lead to increased energy and hardware resource consumption. The instability of STSG and the imperfections of DSR (Meng et al., 2022) motivate us to investigate the following question:

**Question 1:** *How can we maximize the performance of SNN with the freely-optimized potential threshold, while maintaining the training stability and hardware compatibility?*

## 5 PROPOSED METHOD

To answer the question above, we propose *LT-SNN*, a novel SNN training algorithm that successfully resolves the contradiction between threshold optimization, training stability, and hardware compatibility. LT-SNN optimizes the layer-wise potential threshold during training, maximizing the training performance of SNNs without introducing any learning constraints. In the meantime, LT-SNN embraces the advantages of the adaptive potential threshold while maintaining the true binary spikes, bridging the research gap between biological inspiration and practical AI applications.

### 5.1 SEPARATE GRADIENT PATH

As summarized in **Observation 1**, the incompatibility of STSG implies the distinct loss landscape with respect to the gradient descent of the membrane potential and potential threshold. Motivated by that, we propose Separate Gradient Path (SGP), which treats the gradient computation of $u_t$ and $V_{th}$ with dedicated gradient approximations. Specifically, SGP trains SNNs by introducing the Gradient Penalty Window (GPW), a simple-yet-effective method that optimizes the potential threshold without losing training stability. On top of the gradient approximation in Eq. 5, GPW is characterized as a non-linear function $\sigma(\cdot)$, which reshapes the surrogate gradient of the layer-wise potential threshold $V_{th}$. Mathematically, the GPW-aided separate gradient path is characterized as:

$$\frac{\partial S_t}{\partial u_t} = \theta'(u_t - V_{th}) = \max(0, 1 - |u_t - V_{th}|) \tag{8}$$

$$\frac{\partial S_t}{\partial V_{th}} = -\theta'(u_t - V_{th})\sigma(u_t - V_{th}) = -\max(0, 1 - |u_t - V_{th}|)\sigma(u_t - V_{th}) \tag{9}$$

In this work, we choose the Sigmoid function as the gradient penalty window for potential threshold:

$$\sigma(u_t - V_{th}) = \frac{1}{1 + e^{-(u_t - V_{th})}} \tag{10}$$

The choice of Sigmoid function is empirical as it produces the best results among different surrogate functions. Performance comparison of different surrogate functions with LT-SNN is presented in Appendix C, Table 11. For gradient computation of $V_{th}$, we accumulate the gradient computed in Eq. 9 to avoid the dimensionality mismatch:

$$|\frac{\partial L}{\partial V_{th}}| = \frac{\partial L}{\partial S_t}\frac{\partial S_t}{\partial V_{th}} = \frac{\partial L}{\partial S_t} \sum_{(N,C,H,W)} \left( \mathbb{1}\{u_t \geq V_{th}\} \times \theta'(u_t - V_{th})\sigma(u_t - V_{th}) \right) \tag{11}$$

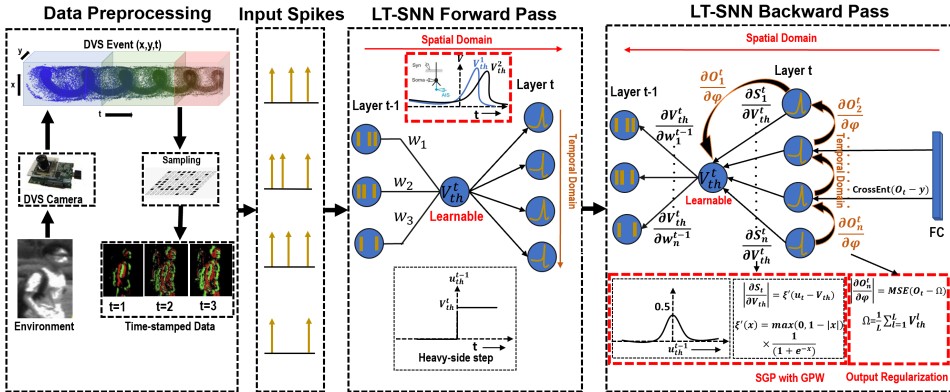

Figure 3: Implementation flow of LT-SNN.

Table 2: Training results of VGG-9 model on DVS-CIFAR10 dataset with different SG schemes.

| Method | Learnable Thre. | SG Func. | Top-1 Accuracy (%) |
|---|---|---|---|
| Fixed threshold Train. | ✗ | Triangle | 78.02 |
| Penalty for all | ✓ | GPW for all | 76.40 |
| **This work** | ✓ | **SGP** | **80.04** |

Since the unfired neurons have no contribution to the final loss, the indicator function $\mathbb{1}\{u_t \geq V_{th}\}$ only keeps the gradient with respect to the active neurons in the forward pass.

We empirically prove the superiority of the proposed SGP algorithm based on different surrogate gradient scenarios. Table 2 summarizes the performance of the directly trained VGG9-SNN on the DVS-CIFAR10 dataset with different gradient computation paths. On top of that, we compare the performance of SGP with non-unified learning rate based LT-SNN training. Table 8 in Appendix C demonstrates the superiority of SGP over non-unified learning rate for LT-SNN. As shown in Table 2, penalizing all the surrogate gradient with GPW leads to sub-optimal performance. On the contrary, the proposed SGP algorithm achieves the best performance with distinct gradient computations. SGP exploits the optimal layer-wise potential threshold with the stabilized training process and fast convergence. Empowered by the proposed SGP scheme, LT-SNN achieves >80% validation accuracy within 50 training epochs, as depicted in Figure 2(b). Compared to the SoTA SNN training with fixed threshold (Deng et al., 2021), SGP embraces the advantage of the adaptive potential threshold learning and achieves superior accuracy. Furthermore, SGP preserves the true binary spikes (0 and 1) in the resultant model, maximizing the hardware compatibility without introducing any high-precision scaling for spike generation (Meng et al., 2022). The complete flow of the proposed SGP with GPW and output regularization is illustrated in Figure 3.

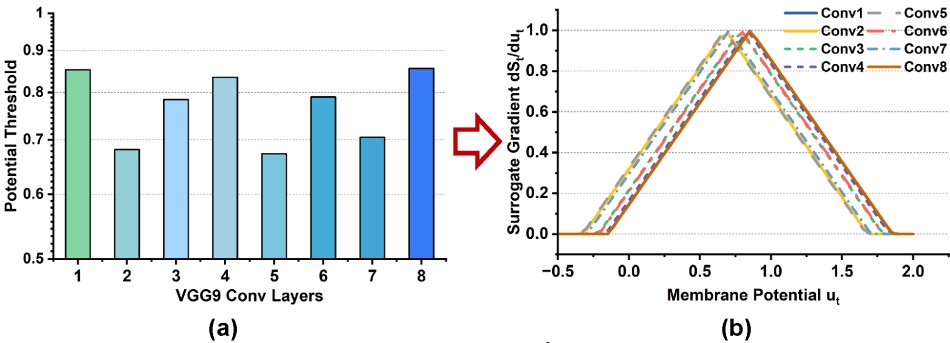

Figure 4: (a) Layer-wise adaptive potential threshold $V_{th}^l$ leads to (b) layer-wise adaptive surrogate gradient gradient of the membrane potential $u_t$.

## 5.2 SELF-ADAPTIVE SNN LEARNING

The proposed SGP algorithm improves the performance of SNN with the unconstrained optimization of the potential threshold. However, the rationale behind the adaptiveness-induced optimal performance remains unclear, which motivates us to investigate the following question:

**Question 2:** *Why the adaptive potential threshold optimizes the performance of SNN model?*

As shown in Eq. 5, the gradient with respect to the membrane potential $u_t$ is approximated based on the surrogate function $\theta'$, which is characterized by the non-linearity of the function and the shifting induced by the threshold $V_{th}$. With the proposed LT-SNN scheme, the adaptive potential threshold leads to the distinct surrogate gradient for membrane potential $u_t$ of each layer, as shown in Figure 4. Compared to SNN training with fixed potential threshold, LT-SNN adjusts the layer-wise gradient information with the adaptive potential threshold.

## 5.3 TEMPORAL AVERAGED LOSS FUNCTION AND OUTPUT REGULARIZATION

We also adopt and customize the temporal efficient training (TET) loss function for output regularization (Deng et al., 2021) to further improve the performance. We adopt the MSE regularization loss from TET to improve the layer-wise spiking activity. On top of that, We have used the average learnable potential threshold of all the layers to compute the MSE loss between output spikes and $V_{th}$. The output of the LT-SNN will be regularized towards the averaged layer-wise potential threshold $\Omega$:

$$L = \beta \frac{1}{T} \underbrace{\sum_{t=1}^{T} L_{CE}(O(t), y)}_{\text{TET Loss}} + (1 - \beta)\frac{1}{T} \underbrace{\sum_{t=1}^{T} \text{MSE}(O(t), \Omega)}_{\text{MSE regularizer}} \tag{12}$$

$$\text{Where} \qquad \Omega = \frac{1}{\ell} \sum_{l=1}^{\ell} V_{th}^l \tag{13}$$

$L_{CE}$ is the cross-entropy loss between the predicted and target class, MSE is the mean square difference between the network output and the regularization target $\Omega$, $O(t)$ represents the output spikes at the final layer, and $T$ represents the simulation time steps of the input sample. While $\beta$ and $\Omega$ are the weights of the moving average and target value of regularization, respectively. We choose $\Omega$ as the mean learnable threshold of the forward pass to regularize the output based on the layer-wise learnable threshold, where $\ell$ represents the total number of layers in our SNN architecture.

## 6 EXPERIMENTAL RESULTS

In this section, we validate the proposed LT-SNN algorithm with multiple event-based computer vision datasets, including DVS-CIFAR10 (Li et al., 2017), N-Cars (Sironi et al., 2018), N-Caltech101 (Orchard et al., 2015) and Prophesee Automotive Gen1 (de Tournemire et al., 2020). Unlike prior works that only employ large-sized VGG or ResNet models (>11M parameters), the proposed algorithm is validated on compact VGG models (2.4-7.1M parameters) and light-weight MobileNet-V1 model (1.5M parameters) (Howard et al., 2017). As mentioned earlier, our objective is to achieve maximum SNN accuracy with the least model parameters. Table 3 summarizes the architectures of the experimented models.

Table 3: Convolutional SNN model architectures for LT-SNN training. "C3", "DW", "MP2" , "AP2" and "FC" represent 3×3 convolution layer, depth-wise separable block (Howard et al., 2017), 2×2 max-pooling, 2×2 average-pooling, and fully connected layer, respectively.

| Model | Architecture |
|---|---|
| MobileNet-Light | 32C3-64DW-64DW-AP2-128DW-128DW-AP2-256DW-AP2-FC256-FC10 |
| VGG-7 | 32C3-32C3-AP2-64C3-64C3-AP2-128C3-128C3-AP2-256C3-256C3-AP2-FC256-FC10 |
| VGG-9 | 32C3-64C3-64C3-AP2-128C3-128C3-AP2-256C3-256C3-AP2-512C3-512C3-AP2-FC512-FC10 |
| Yolo-V2 | 32C3-MP2-64C3-MP2-128C3-64C1-128C3-MP2-256C3-128C1-256C3-MP2-512C3-256C1-512C3-256C1-MP2-1024C3-512C1-1024C3-AP2-FC512-FC576 |

Table 4: Experimental results of the proposed LT-SNN on DVS-CIFAR10 and CIFAR-10 datasets. Except "VGG-9 (4-bit)", 32-bit precision is used for all results of prior works and this work.

| Dataset | Method | Architecture | Parameters | Simulation Length | Accuracy(%) |
|---|---|---|---|---|---|
| DVS-CIFAR-10 | ASF-BP (Wu et al., 2021) | VGG-Like | 15.2M | 50 | 62.50 |
| | tdBN (Zheng et al., 2021) | ResNet-19 | 11.18M | 10 | 67.80 |
| | ParamLIF (Fang et al., 2021b) | VGG-Like | 17.4M | 20 | 74.80 |
| | RecDis (Guo et al., 2022) | ResNet-19 | 11.18M | 10 | 72.42 |
| | TET (Deng et al., 2021) | VGG-Like | 9.27M | 10 | 77.33 |
| | DSR (Meng et al., 2022) | VGG-11 | 9.34M | 30 | 75.70 |
| | **This work** | VGG-11 | **9.34M** | **30** | **79.51** |
| | **This work** | MobileNet-V1 (light) | **1.28M** | **30** | **75.70** |
| | **This work** | VGG-7 | **1.91M** | **30** | **78.93** |
| | **This work** | VGG-9 (4-bit) | **7.07M** | **30** | **80.04** |
| | **This work** | VGG-9 | **7.07M** | **10** | **79.10** |
| | **This work** | VGG-9 | **7.07M** | **8** | **78.30** |
| CIFAR-10 | Hybrid (Rathi & Roy, 2020) | ResNet-20 | 12.3M | 10 | 92.54 |
| | tdBN (Zheng et al., 2021) | ResNet-19 | 11.18M | 6 | 93.16 |
| | DSR (Meng et al., 2022)[1] | ResNet-18 | 11.18M | 6 | 91.89 |
| | **This work** | ResNet-19 | **11.18M** | **6** | **93.97** |

[1] The experiment is rigorously performed based on the open-sourced DSR implementation with 6 time steps.

Table 5: Experimental results of the proposed LT-SNN on N-Caltech101 and N-Cars datasets.

| Dataset | Method | Representation | Accuracy(%) | Pretrain |
|---|---|---|---|---|
| N-Caltech101 | YOLE (Cannici et al., 2019) | Histogram | 70.02 | False |
| | EST (Gehrig et al., 2019) | Histogram | 81.70 | True |
| | AsyNet (Messikommer et al., 2020) | Histogram | 76.10 | False |
| | AEGNN (Schaefer et al., 2022) | Graph | 66.80 | False |
| | **This work** | **Binary Spike** | **77.71** | **False** |
| N-Cars | NDA (Li et al., 2022) | Binary Spike | 91.9 | False |
| | AsyNet (Messikommer et al., 2020) | Histogram | 94.4 | False |
| | YOLE (Cannici et al., 2019) | Histogram | 92.7 | False |
| | AEGNN (Schaefer et al., 2022) | Graph | 94.5 | False |
| | Object (Cordone et al., 2022) | Binary Spikes | 92.4 | False |
| | **This work** | **Binary Spike** | **95.02** | **False** |

**SNN Training for Full-/Low-Precision Inference**    We perform LT-SNN training to execute inference with both full-precision and 4-bit precision. To train SNNs for low-precision inference, we adopt the Statistic-Aware Weight Binning (SAWB) quantizer (Choi et al., 2019) to compress the layer-wise weights of the LT-SNN-VGG-9 network down to 4-bit precision. We separately perform and compare the full-precision and quantized-aware training of our SNN architectures using hyperparameters mentioned in Appendix A. Using our LT-SNN scheme, we achieve SoTA results for both full-precision and low-precision SNNs, as reported in Table 4 and Table 5. Proposed LT-SNN and DSR train the threshold in a layer-wise manner and the potential threshold for each layer is different. The rest of the results reported from other SoTA employ fixed $V_{th}$ for all the layers.

**Image Classification Task**    We validate the proposed LT-SNN algorithm on both event-based image classification datasets and conventional computer vision dataset with RGB images. The LT-SNN models are directly trained from scratch, and the detailed input preprocessing and experimental setup are summarized in Appendix A. Table 4 summarizes the performance of LT-SNN on DVS-CIFAR10 and CIFAR-10 (with RGB images) datasets. For DVS-CIFAR10 dataset, compared to the current SoTA method (Deng et al., 2021), the 4-bit VGG-9 model trained by proposed LT-SNN algorithm achieves 2.71% accuracy improvement with $1.31\times$ less parameters and $10.48\times$ model size reduction (MB). Furthermore, the proposed LT-SNN demonstrates consistently superior performance with different simulation length (from 8 time steps to 30 time steps). For the conventional RGB CIFAR-10 dataset, LT-SNN also achieves new SoTA performance with ResNet-19 architecture, where we adopted the tdBN (Zheng et al., 2021) scheme to stabilize the direct SNN training process. Based on the official implementation of DSR (Meng et al., 2022), we rigorously conduct the experiment of DSR on CIFAR-10 dataset with 6 simulated time steps. Additionally, to demonstrate the effectiveness of learnable threshold, we conduct more experiments using SGP with and without learnable threshold. Proposed LT-SNN with SGP surpassed SNN implementations with the fixed threshold for different compact networks reported in Appendix C, Table 9.

In addition to the CIFAR datasets (Li et al., 2017), we evaluate the proposed LT-SNN algorithm on N-CalTech101 (Orchard et al., 2015) and N-Cars (Sironi et al., 2018) datasets. As shown in

Table 5, the proposed LT-SNN achieves new SoTA performance on both datasets with 1.61% (N-Caltech101, without pre-training) and 0.52% (N-Cars) accuracy improvements. Compared to the conventional histogram-based computation (Messikommer et al., 2020; Cannici et al., 2019) that use non-binary activations, the proposed LT-SNN enables end-to-end binary computation with binary spikes throughout the entire SNN, achieving superior performance with high hardware compatibility. Additional results comparing the baseline of LT-SNN with TET and DSR using DVS-CIFAR10 dataset are summarized in Appendix C, Table 7.

**Impact of the Event Time Steps** SNNs require iterative computation and membrane potential accumulation, which motivated prior works to exploit the computation reduction with less number of time steps to represent the incoming event. We also validated LT-SNN with different simulated time steps on the DVS-CIFAR10 dataset. As shown in Table 4, the proposed LT-SNN manages to achieve SoTA performance with reduced time steps and compact models, compared to the prior works. With the same 10 time steps as TET Deng et al. (2021), LT-SNN achieves 1.77% accuracy improvement, without using any data augmentation. With the extended 30 time steps, LT-SNN achieves 80.04% SoTA accuracy on DVS-CIFAR10 dataset.

**Object Detection Task** In addition to the classification datasets, we validate the proposed LT-SNN on large-sized Prophesee Automotive Gen1 dataset (de Tournemire et al., 2020) for object detection task. With 228,123 bounding boxes for cars and 27,658 for pedestrians, the Gen1 dataset (de Tournemire et al., 2020) is considered as the most complex event-based computer vision task.

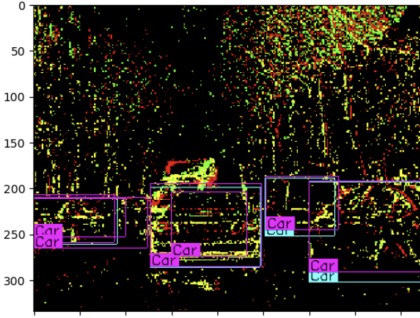

Figure 5: Inference results of LT-SNN on Prophesee Automotive Gen1 dataset.

Unlike the prior works use accumulated histogram (Messikommer et al., 2020; Cannici et al., 2020; Cordone et al., 2022) or graph-based input representation (Schaefer et al., 2022), we translate DVS event to pure binary frames and synchronize them with artificial actual ground truths from (Perot et al., 2020) for network training. The detailed data preprocessing setup is summarized in Appendix A. The binary input events and intermediate spikes enables the end-to-end binarized computing for LT-SNN, elevates the computation efficiency with simplified data format. We develop LT-SNN-YoloV2 feature extraction network followed by Yolo loss to localize the objects and compute mean average precision (mAP) for detected bounding boxes against the ground truths as shown in Figure 5. Table 6 and Table 10 of Appendix C elucidate that, in comparison to the current SNN-based SoTA with fixed potential threshold, our LT-SNN-custom-Yolov2 records SoTA mAP of 0.298 on Prophesee Automotive Gen1 dataset.

Table 6: Experimental results of the proposed LT-SNN on Prophesee Automotive Gen1 dataset.

| Method | Model Architecture | Spiking Model | mAP |
|---|---|---|---|
| Asynet (Messikommer et al., 2020) | FB-Sparse | No | 0.145 |
| MatrixLSTM (Cannici et al., 2020) | ResNet-19 | No | 0.3 |
| RED (Perot et al., 2020) | RetinaNet | No | 0.41 |
| VGG-11+SSD (Cordone et al., 2022) | **VGG+SSD-SNN** | Yes | 0.187 |
| **This work** | **Custom-YoloV2-SNN** | **Yes** | **0.298** |

## 7 CONCLUSION

In this work, we propose a novel SNN training algorithm with learnable threshold (LT-SNN), which optimizes the layer-wise threshold with direct SNN training. As one of the first studies on this topic, the proposed LT-SNN unleashes the firing constraints that were imposed in prior works, without introducing any additional high-precision scaling factor to the spike generation. The proposed method with full and low-precision training has been verified on multiple event-based image classification and object detection datasets. LT-SNN improves the state-of-the-art accuracy for DVS-CIFAR10 dataset by 2.8% together with 10.48× smaller model size. For object detection on Prophesee Automotive Gen1 dataset, the LT-SNN outperforms SNN-based SoTA mAP by 0.11 with end-to-end binary computation.

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

# A  DETAILED EXPERIMENTAL SETUP

**Data Preprocessing**  To ensure hardware efficient LT-SNN based training, we convert DVS-events to binary frames in the event pre-processing stage. For classification data, we perform sampling over different time steps and transform events to binary tensors of shape [Batch, Time, Channels, Height, Width]. Similarly, we follow the same convention to convert Prophesee Gen1 events to binary frames. Then the generated binary frames are synchronized with artificial ground truth from (Perot et al., 2020). Finally, the events and annotations are translated to the tensors of shape [Batch, Time, Channels, Height, Width] and [Batch, Number of boxes, Bounding box] respectively.

**Training Parameters**  We train our proposed LT-SNN based classification and object detection architectures using PyTorch. Regarding hyperparameter selection, we use the Adam optimizer where the learning rate is set to 0.001 and cross entropy loss plus mean square error is computed between detected and target classes for back-propagation and output regularization. For output regularization, the moving average weight $\beta$ is set to 0.45 for both the full-precision and low-precision training. Finally, each LT-SNN based architecture from Table 3 was trained for 200 epochs and we observe robust convergence of our proposed LT-SNN based architectures in less than 50 epochs.

# B ALGORITHM

---

**Algorithm 1** LT-SNN Algorithm

---

**Initialize:** Input spike as $S_t^0$ ; Membrane potential as $u_t^n$ ; Learnable potential threshold as $V_{th}^0$;
Output spike as $O_t^0$ and regularization ratio as $\beta$
**for** $t \leftarrow 1$ *to* $T$ **do**
 **Forward Pass:**
 Compute $u_t^1, O_t^1$ using Eq. 2
 **for** $n \leftarrow 1$ *to* $N$ **do**
  **Forward Pass:**
  Compute $u_t^n, O_t^n$ using Eq. 3
 **end**
**end**
$p = \text{Prediction}(Y, O_t^n)$
**Loss:**
Compute $\Omega$ using Eq. 13 with detached $V_{th}$.
$L_{LT\_SNN} = L_{CE}(Y, O_t^n) + (1 - \beta)MSE(O_t^n, \Omega)$
**Backward Pass:**
**Initialize:**
$\partial L_{LT\_SNN} / \partial O_t^n \leftarrow 0$
**for** $t \leftarrow T$ *to* $1$ **do**
 $\partial L_{LT\_SNN} / \partial O_t^n = \text{Spatial\_Gradient}(\partial L_{CE} / \partial O_t^n) + \text{Output\_Regularization}(\partial MSE / \partial O_t^n)$
 **for** $n \leftarrow N$ *to* $1$ **do**
  Compute gradient using Eq. 6
  Compute $\partial L_{CE} / \partial V_{th}^n$ using Eq. 9 and Eq. 10
  Mask $\partial L_{CE} / \partial V_{th}^n$ with active pixel density using Eq. 11
 **end**
**end**

---

## C    ADDITIONAL EXPERIMENTAL RESULTS

we have trained the VGG-11 architecture reported by DSR and TET using the proposed SGP for the layer-wise learnable threshold to present an apple-to-apple comparison of LT-SNN with DSR and TET work. Table 4 elucidates that LT-SNN shows 3.81% and 2.18% accuracy improvement in comparison to the DSR and TET respectively.

Table 7: Baseline comparison of proposed SGP with current SoTA.

| Method | Architecture | Threshold | SG Func. | Top-1 Accuracy (%) |
|---|---|---|---|---|
| TET [R3] | VGG-11 | Fixed | Triangle | 77.33 |
| DSR [R4] | VGG-11 | Fixed | STSG | 75.70 |
| **This work** | **VGG-11** | **Learnable** | **SGP** | **79.51** |

To compare the performance of SGP with lower learning rate based SNN training, we have performed extra experiments with VGG-9 using high lr (0.1) for $W$ and relatively low lr (0.001) for $V_{th}$. We observe that such high lr value for $W$ with "Adam" optimizer didn't help in model convergence. We performed another experiment with lr (0.001) for $W$ and lr (0.00001) for $V_{th}$. The Table 8 summarizes the experimental results, which show that the separate learning rate setting is not sufficient to fully optimize the learnable potential threshold.

Table 8: Performance comparison of SGP and lower learning rate to balance out $V_{th}$ and $W_{ij}$ for LT-SNN training.

| Model | W lr | Vth lr | SGP | Accuracy (%) | Converged |
|---|---|---|---|---|---|
| VGG-9 | 0.1 | 0.001 | ✗ | 20.40 | ✗ |
| VGG-9 | 0.001 | 0.00001 | ✗ | 77.20 | ✓ |
| **VGG-9** | **0.001** | **0.001** | **✓** | **80.04** | **✓** |

We have conducted more experiments by disabling the learnable threshold to compare the performance of fixed and learnable threshold with the identical setting. We have used DVS-CIFAR10 data for the classification task and Prophesee Automotive Gen1 data for the object detection task. Table 9 and Table 10 demonstrate the superiority of our proposed layer-wise learnable potential threshold scheme in comparison to the fixed potential threshold for all the layers. Proposed SGP for learnable potential threshold increases SoTA accuracies by 2.02% and 1% with VGG-9 and MobileNet-v1 respectively and mAP by 0.176 with Custom-YoloV2-SNN.

Table 9: Performance comparison of learnable potential threshold and fixed potential threshold for the classification networks.

| Method | Architecture | Threshold | SG Func. | Top-1 Accuracy (%) |
|---|---|---|---|---|
| This work | VGG-9 | Fixed | Triangle | 78.02 |
| **This work** | **VGG-9** | **Learnable** | **SGP** | **80.04** |
| This work | MobileNet-v1(Lite) | Fixed | Triangle | 74.70 |
| **This work** | **MobileNet-v1(Lite)** | **Learnable** | **SGP** | **75.70** |

Table 10: Performance comparison of learnable potential threshold and fixed potential threshold for the object detection networks.

| Method | Architecture | Threshold | SG Func. | mAP |
|---|---|---|---|---|
| **This work** | Custom-YoloV2-SNN | Fixed | Triangle | 0.122 |
| **This work** | **Custom-YoloV2-SNN** | **Learnable** | **SGP** | **0.298** |

We choose the sigmoid function for SGP based on the experimental results that we achieved by setting up an ablation study. From Table 11, we could conclude that the sigmoid function is the best choice to ensure stable training performance and high validation accuracy.

Table 11: Comparison of different surrogate functions performance as SGP.

| Model | Epochs | SGP | Accuracy (%) | Converged |
|-------|--------|-----|--------------|-----------|
| VGG-9 | 200 | ArcTan | 77.81 | ✓ |
| VGG-9 | 200 | Triangle | 70.83 | ✓ |
| VGG-9 | 200 | Piece-wise | 78.81 | ✓ |
| **VGG-9** | **200** | **Sigmoid** | **80.04** | ✓ |

To ensure the training stability of the proposed SGP, we have trained VGG-9 for 200 epochs. From Figure 6, we can observe that SGP for LT-SNN didn't hurt the training stability for 200 epochs.

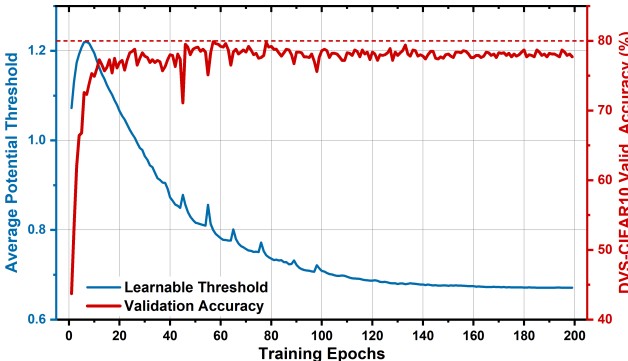

Figure 6: Stable training process of the proposed LT-SNN algorithm with the extended training effort on DVS-CIFAR10 dataset.

