# OpenReview forum: "LT-SNN: Self-Adaptive Spiking Neural Network for Event-based Classification and Object Detection"
_ICLR.cc/2023/Conference — Submitted to ICLR 2023_

### Official Review · Reviewer_A7Jv · 2022-10-24

**Confidence:** 2
**Correctness:** 4
**Technical Novelty And Significance:** 3
**Empirical Novelty And Significance:** 3
**Recommendation:** 5

**Clarity, Quality, Novelty And Reproducibility:**

Overall, the paper itself is well written, and the simplicity of the method lends itself to ease of replicability. Overall, the work seems original, and the results are impressive.

**Strength And Weaknesses:**

Overall, the motivation and method is very intuitive, and the proposed solution is simple and makes sense. The authors provide a good amount of background, and justify well the need for the proposed contribution. While the method is quite simple, the improvements in the experiments are quite impressive, especially considering the significant reduction in model size.

One open question is how much the regularization in Section 5.3 contributes to the work. It seems to only be briefly mentioned, but is not ablated, and it's unclear how important this is to the improvements seen in the experiments.

**Summary Of The Paper:**

This paper presents a novel training method for Spiking Neural Networks. Typically, the non-differentiability of the spiking neuron was by-passed with a differentiable approximation (e.g. the triangle function). However, each spike depends on two variables: the previous neuron's voltage, and a voltage threshold above which the spike is created. In prior works, the network optimized over the voltage during training, but kept the voltage threshold as a constant, due to training instability issues. This work resolves these instability issues by introducing a separate gradient approximation for the voltage threshold, consisting of the triangle function modulated by a sigmoid function. During backpropagation, the voltage and voltage threshold will receive separate gradients based on this method. Empirical experiments are performed to demonstrate the stability of this method, and the authors demonstrate significant improvements in experiments for image classification and object detection over prior works, with much smaller model sizes.

**Summary Of The Review:**

Overall, the proposed method seems like a simple modification to existing SNN networks which provides a significant improvement. However, I may be missing some context as to the novelty of this work.

EDIT: After reviewing the discussions from the other reviewers, it seems like there are legitimate concerns about the novelty and experiments of this work. Taking this in mind, I am modifying my review to a 5 (although more of a 5.5).

---

> ### Author Response · Authors · 2022-11-19
> **Response to Reviewer A7Jv: Q1**
>
> **Q1**: One open question is how much the regularization in Section 5.3 contributes to the work. It seems to only be briefly mentioned, but is not ablated, and it's unclear how important this is to the improvements seen in the experiments.
>
> **A1**: Thank you for raising this question. MSE regularizer is incorporated to maximize the spike activity based on the optimized learnable threshold at the final layer. As mentioned in the original manuscript, we adopt MSE regularization loss from [R1]. On top of that, We have used the average learnable potential threshold of all the layers to compute the MSE loss between output spikes and $V_{th}$. Following the reviewer’s suggestion, we updated Section 5.3 in the revised manuscript with the clarified description of the loss computing. Empirically, our LT-SNN scheme achieved 2.71% higher accuracy compared to TET.

---

> > ### Author Response · Authors · 2022-11-19
> > **References**
> >
> > **References**
> >
> > [R1] Deng et al. "Temporal Efficient Training of Spiking Neural Network via Gradient Re-weighting." In International Conference on Learning Representation(ICLR),2022.
> >
> > [R2] Zheng et al. ”Going deeper with directly-trained larger spiking neural networks”. In Proceedings of the AAAI Conference on Artificial Intelligence, 2021.
> >
> > [R3] Fang et al. ”Deep residual learning in spiking neural networks.” In Advances in Neural Information Processing Systems, 2021.
> >
> > [R4] Mozzachiodi et al. ”More than synaptic plasticity: role of nonsynaptic plasticity in learning and memory.” In Trends in neurosciences, 2010.
> >
> > [R5] Azouz et al. ”Dynamic spike threshold reveals a mechanism for synaptic coincidence detection in cortical neurons in vivo.” In Proceedings of the National Academy of Sciences, 2000.
> >
> > [R6] Meng et al. ”Training High-Performance Low-Latency Spiking Neural Networks by Differentiation on Spike Representation.” In Proceedings of the IEEE/CVF Conference on Computer Vision and Pattern Recognition(CVPR), 2022.
> >
> > [R7] Cordone et al. ”Object Detection with Spiking Neural Networks on Automotive Event Data.” In Proceedings of the IEEE/CVF Conference on Computer Vision and Pattern Recognition(CVPR), 2022.
> >
> > [R8] Maarten H, et al. ”Is Action Potential Threshold Lowest in the Axon?” Nature Neuroscience, 11(11):1253–1255, 2008.

---

> > > ### Comment · Reviewer_A7Jv · 2022-11-21
> > > **Thanks to the authors for the responses**
> > >
> > > After reviewing the discussions from the other reviewers, it seems like there are legitimate concerns about the novelty and experiments of this work. Taking this in mind, I am modifying my review to a 5 (although more of a 5.5).

---

> > > > ### Author Response · Authors · 2022-12-10
> > > > **Response to Reviewer A7Jv**
> > > >
> > > > We addressed all the questions from the reviewers and John. We hope our detailed responses can resolve all the questions and concerns, please kindly reconsider the overall score of our work.
> > > > Thanks
> > > >
> > > > Best,
> > > > Authors

---

### Official Review · Reviewer_fvZZ · 2022-10-25

**Confidence:** 4
**Correctness:** 4
**Technical Novelty And Significance:** 2
**Empirical Novelty And Significance:** 2
**Recommendation:** 5

**Clarity, Quality, Novelty And Reproducibility:**

Clarity:
The paper is clearly written.

Quality:
The experiments show good results. But the problem importance and  the motivation of the proposed method is missing.

Novelty:
The novelty is trivial.

Reproducibility:
The authors fail to provide code.

**Strength And Weaknesses:**

Strength

1. The proposed method is shown to outperform SOTA on multiple event-based datasets.
2. The paper is well written and easy to follow.


Weakness

1. I don't see a clarification for the problem importance. Why learnable threshold is important in SNN? If the authors want to make SNN more biologically plausible, should the threshold lie in the specific range? I don't see any discussion or insight in the paper.
2. The proposed method is not well motivated. For example, the authors propose a separate gradient path in sec. 5.1, but why we should use a separate path instead of single one? Is there any insight or theoretical guarantee? Also, for Eq.9, the authors select the sigmoid function as the activation. Why would the authors select this particular activation function? Is there any biologically insight?

**Summary Of The Paper:**

This paper propose LT-SNN, a novel SNN training algorithm with self-adaptive learnable potential threshold to improve SNN performance. To stabilize the SNN training even further, the author propose separate surrogate gradient path (SGP), a simple-yet-effective method that enables the smooth learning process of SNN training. The proposed method is validate on multiple event-based datasets, including both image classification and object detection tasks.

**Summary Of The Review:**

Although the paper gets better results with more parameters, the problem importance and the motivation of the proposed method is unclear. Therefore, I give a weak reject for the paper.

---

> ### Author Response · Authors · 2022-11-19
> **Response to Reviewer fvZZ: Major Q1:Q2**
>
> **Q1**: I don't see a clarification for the problem importance. Why learnable threshold is important in SNN? If the authors want to make SNN more biologically plausible, should the threshold lie in the specific range? I don't see any discussion or insight in the paper.
>
> **A1**: Thank you for raising this question. The fundamental problem of importance for us is to improve the accuracy of SNNs with smaller model sizes. To achieve this, we make observations that the fixed threshold scheme in prior SNN algorithms ignores the activity difference among different layers, limiting the learnability and accuracy. Inspired by the literature on biological nervous systems with dynamic neuronal thresholds, the proposed LT-SNN exploits the layer-wise adaptiveness of SNN by learning the adaptive firing threshold values. We empirically show that this enhanced the learnability and considerably improved the SNN accuracy. Since our goal is achieving higher SNN accuracy rather than only making SNNs more biologically plausible, we encourage the model to learn the threshold rather than constraining the optimization in a specific range.
>
> We have made several modifications in Section 1 of the revised manuscript to clarify the biological inspiration and our objective of improving the SNN accuracy. We also removed a few occurrences of ambiguous biological plausibility.
>
> **Q2**: The proposed method is not well motivated. For example, the authors propose a separate gradient path in sec. 5.1, but why we should use a separate path instead of single one? Is there any insight or theoretical guarantee? Also, for Eq.9, the authors select the sigmoid function as the activation. Why would the authors select this particular activation function? Is there any biologically insight?
>
> **A2**: As mentioned in Section 4, paragraphs "Hypothesis 1" and "Observation 1", both $\partial S/\partial u$ and $\partial S/\partial V_{th}$ are two different landscapes and are not directly transferable with straight-through surrogate gradient (STSG). Therefore, to achieve training stability with maximum performance, we optimized the layer-wise learnable potential threshold with the proposed separate gradient path (SGP) consisting of the sigmoid-based gradient penalty window (GPW).
> We have empirically validated the significance of our separate gradient path (SGP) approach in comparison to the straight-through surrogate gradient (STSG) to achieve high training stability in the deep SNN networks. The performance comparison of both approaches is explicit in Figure 2.
> The choice of sigmoid function for SGP is based on the experimental results that we achieved by setting up the ablation study. From the table below, we could conclude that the sigmoid function is the best choice to ensure stable training performance and high validation accuracy. We have included the table below in the Appendix section of the revised manuscript as Table 11.
>
> *[Table: Comparison of different surrogate functions' performance as SGP.]*
>
> | **Model** | **Epochs** | **SGP**     | **Accuracy** | **Converged** |
> | --------- | ---------- | ----------- | ------------ | ------------- |
> | VGG-9     | 200        | ArcTan      | 77.81        | Yes           |
> | VGG-9     | 200        | Triangle    | 70.83        | Yes           |
> | VGG-9     | 200        | Piece-wise  | 78.81        | Yes           |
> | **VGG-9** | **200**    | **Sigmoid** | **80.04**    | **Yes**       |

---

> > ### Comment · Reviewer_fvZZ · 2022-11-20
> > **Thanks for your response.**
> >
> > I don't think the authors have successfully address my concern.
> >
> > The author claim to "improve the accuracy of SNNs with smaller model sizes", but fail to give biological insight. Only showing improvement in numbers makes it unclear to know whether the improvements result from adding additional trainable parameters.
> >
> > Furthermore, the proposed algorithm is not well biologically motivated, either.
> >
> > Therefore, I still to my original score.

---

> > > ### Author Response · Authors · 2022-12-10
> > > **Response to Reviewer fvZZ Part 1**
> > >
> > > ```markdown
> > > I don't think the authors have successfully address my concern.
> > > The author claim to "improve the accuracy of SNNs with smaller model sizes", but fail to give biological insight. Only showing improvement in numbers makes it unclear to know whether the improvements result from adding additional trainable parameters.
> > > Furthermore, the proposed algorithm is not well biologically motivated, either.
> > >
> > > Therefore, I still to my original score.
> > > ```
> > >
> > > Thank you for your response. We respectfully disagree with your comments based on the following reasons:
> > >
> > > 1.  We conducted additional ablation study experiments by disabling the learnable threshold while keeping the other settings to be identical. In other words, our objective is to demonstrate the effectiveness of introducing the learnable threshold into the training process. The following table summarizes the classification results on DVS-CIFAR10 dataset with VGG-9 and MobileNetV1 (Lite) architectures:
> > >
> > >    *[Table: Performance comparison of learnable potential threshold and fixed potential threshold for the classification networks.]*
> > >
> > >    | **Method**    | **Architecture**       | **Threshold** | **SG Func.** | **Top-1 Accuracy** |
> > >    | ------------- | ---------------------- | ------------- | ------------ | ------------------ |
> > >    | This work     | VGG-9                  | Fixed         | Triangle     | 78.02              |
> > >    | **This work** | **VGG-9**              | **Learnable** | **SGP**      | **80.04**          |
> > >    | This work     | MobileNet-v1(Lite)     | Fixed         | Triangle     | 74.70              |
> > >    | **This work** | **MobileNet-v1(Lite)** | **Learnable** | **SGP**      | **75.70**          |
> > >
> > >    In addition to the DVS-CIFAR10 dataset, we validated the performance of LT-SNN on Prophesee Gen1 dataset with Customized-YOLOV2-SNN:
> > >
> > >    *[Table: Performance comparison of learnable potential threshold and fixed potential threshold for the object detection networks.]*
> > >
> > >    | **Method**    | **Architecture**      | **Threshold** | **SG Func.** | **mAP**   |
> > >    | ------------- | --------------------- | ------------- | ------------ | --------- |
> > >    | This work     | custom-Yolov2-SNN     | Fixed         | Triangle     | 0.122     |
> > >    | **This work** | **custom-Yolov2-SNN** | **Learnable** | **SGP**      | **0.298** |
> > >
> > >
> > > Both of the above tables are added to the Appendix of the revised manuscript as Table 9 and Table 10. The empirical results in the above tables justify the superior performance of the proposed LT-SNN algorithm compared to the fixed potential threshold learning.
> > >
> > >  Introducing learnable potential threshold resulted in the elevated performance among various spiking neural network architectures against multiple datasets. The experimental results successfully demonstrate the noticeable improvements of introducing the additional learnable potential threshold into training.

---

> > > > ### Author Response · Authors · 2022-12-10
> > > > **Response to Reviewer fvZZ Part 2**
> > > >
> > > > 2. Regarding biological motivation, spiking neural network (SNN) is originally motivated by the membrane potential accumulation and the firing procedure of the nervous system. Compared to the actual nervous system, the gradient-based optimization procedure of SNNs is largely different from the memorization mechanism of the human brain. We would like to highlight the fact that the success of SNN is inspired by the information propagation procedure between neurons, rather than exactly mimicking the actual neural system.
> > > >
> > > >    Ion channels of neurons vary inside the membranes of different neurons, which further leads to different thresholds, excitability, and firing patterns for neurons, as reported in [R1]. However, the conventional biology-inspired SNN research works did not incorporate such an adaptive firing threshold scheme. On the algorithm level, associating every single receptive field (neurons) of SNN with a corresponding threshold value will largely increase the computation complexity and memory consumption of SNN, especially with efficiency-oriented AI applications. In our paper, we proposed the layer-wise adaptive threshold scheme, which balances the original biological inspiration and practical algorithmic application. We agree with the reviewer that the inspiration from the biological system provides powerful guidance for algorithm design, and this certainly was the case for our proposed LT-SNN, but at the same time, the proposed algorithm should show noticeable accuracy improvement in numbers on well-known datasets.
> > > >
> > > >    As demonstrated in the responses to reviewer bjTV (Answer for concern 1 and concern 2), only introducing the additional trainable threshold $V_{th}$ parameter cannot lead to optimal overall performance, despite the fine-grained learning rate tuning and different threshold initialization experiments that we performed. The reported experiment results are copied here too, for your convenience.
> > > >
> > > >    *[Table: Performance comparison on DVS-CIFAR10 dataset with different learning rates and initial potential threshold values.]*
> > > >
> > > > | **Architecture** | **Lr-W** | **Lr-Vth** | **Initial Vth** | **SGP** | **Top-1 Accuracy (%)** |
> > > > |------------------|----------|------------|-----------------|---------|------------------------|
> > > > | VGG-9            | 0.1      | 0.001      | 1               | No      | 20.40                  |
> > > > | VGG-9            | 0.001    | 0.00001    | 1               | No      | 77.20                  |
> > > > | VGG-9            | 0.001    | 0.0001     | 1               | No      | 78.35                  |
> > > > | VGG-9            | 0.001    | 0.00025    | 1               | No      | 76.50                  |
> > > > | VGG-9            | 0.001    | 0.0005     | 1               | No      | 78.60                  |
> > > > | VGG-9            | 0.001    | 0.0005     | 2               | No      | 77.80                  |
> > > > | VGG-9            | 0.001    | 0.0005     | 4               | No      | 77.40                  |
> > > > | VGG-9            | 0.001    | 0.0005     | 6               | No      | 72.70                  |
> > > > | **VGG-9**        |**0.001** |**0.001**   | **1**           | **Yes** | **80.07**              |
> > > >
> > > > The proposed LT-SNN algorithm shows the optimal model performance with the proposed separate gradient paths (SGP) and penalty window (GPW), the superior performance of LT-SNN proves the fact that na\"{\i}vely adding the learnable threshold as training parameters is not sufficient for model optimization.
> > > >
> > > > As we addressed all the questions and concerns raised by you as well as other reviewers, please kindly reconsider your decision regarding our submission.
> > > >
> > > > [R1] Sardi, Shira, et al. "New types of experiments reveal that a neuron functions as multiple independent threshold units." *Nature Scientific reports* 7.1 (2017): 1-17.

---

### Official Review · Reviewer_Dob4 · 2022-10-25

**Confidence:** 5
**Correctness:** 2
**Technical Novelty And Significance:** 1
**Empirical Novelty And Significance:** 1
**Recommendation:** 3

**Clarity, Quality, Novelty And Reproducibility:**

The motivation for the work is very unclear to me at the moment. In addition, I did not see much novelty either. Writing needs significant improve!

**Strength And Weaknesses:**


Weaknesses and questions:

-1. The experiments recorded in Table 1 are not clear to me.

       a. Why do "such constraints in the training process largely limits the learnability of SNN"? What's the definition of "learnability"?
       b. What is the definition of "true binary spikes"? Biologically, an action potential is triggered when the membrane potential is higher than a threshold. The output potential should NOT be a binary value in any case. The spike (action potential) is binary.
       c. How hardware deals with layer-wise varied spikes is out of the scope of the paper. I assume the "hardware" means neuromorphic hardware. Biologically, each neuron has a different dynamic threshold. The current neuromorphic hardware can barely mimic the simplest biological neuron. Thus, algorithm and theoretical study should not be limited or favored to existing neuromorphic hardware.
       d. DSR and LTSNN used different base models, VGG-11 for DSR and VGG-9 for LTSNN. Is it fair?

-2. The falsified hypothesis 1 is based only on VGG-9 and DVS-CIFA10 is not convincing. In other words, how general of observation 1? In addition, I am not convinced by the experiments associated with Figure 2. If we stop training STSG before 80 epoch, STSG also has stabilized training, right? In other words, why the authors stopped training LTSNN at epoch 100? What would happen if the authors train LTSNN 200 epoch?

 -3. Super unclear of section 5.3! The L in Eq. 12 is the same as the one in Eq. 13? What's L_{CE}, MSE, O(t), and T? More importantly, why the MSE regularizer is necessary?

-4. For the experiments associated with Table 4, why did the authors not use the same architecture of other competing approaches? What's the logic behind the experimental design?

**Summary Of The Paper:**

The authors proposed a layer-wise learnable threshold approach for SNNs. The proposed method design two separate surrogate gradient paths for the gradient of membrane potential and learnable threshold. Different classification and object detection tasks reflect the effectivenesses of the proposed LTSNN training scheme.

**Summary Of The Review:**

Too many things are unclear to me at the moment, as discussed in the section on "strengths and weaknesses." In addition, many works have developed different dynamic threshold schemes for SNNs. The authors did not mention them at all. For example, in "Biologically Inspired Dynamic Thresholds for Spiking Neural Networks," the authors define dynamic thresholds without learning. More importantly, I do not think the proposed approach is able to imitate "the self-adaptiveness of the biological nervous system."

In summary, I do not recommend the submission.

---

> ### Author Response · Authors · 2022-11-19
> **Response to Reviewer Dob4: Q1**
>
> **Q1**: The experiments recorded in Table 1 are not clear to me.
>
> - **A**: Why do ``such constraints in the training process largely limits the learnability of SNN"? What's the definition of "learnability"?
>
>   - **Ans A**: Thank you for your question. As shown in Eq. 7 of the original manuscript, the spikes generated by DSR are multiplied with the learnable potential threshold $V_{th}$. Therefore, the value of the resultant spikes is $V_{th}$. In the meantime, the actual threshold value of DSR is $\alpha \times V_{th}$, which means the generated spikes of the DSR is always $1/\alpha$ times the learnable threshold $V_{th}$. For instance, $\alpha=0.3$ means the fired spike is always 3.33$\times$ of the actual potential threshold, regardless of the value of $V_{th}$. However, the performance of such constrained firing process is largely impacted by the value of $\alpha$. When $\alpha=1.0$, it means that both the threshold and spikes are freely learned during training, but simultaneously learning spiking magnitude and potential threshold leads to the sub-optimal performance, as shown in Table 1 of the original manuscript. Therefore, we define the sub-optimal learnability of DSR based on the fact that the performance is largely depending on the heuristic selection of $\alpha$, rather than relying on the network training itself. The implementation of Eq. 7 (of the original manuscript) can be found in the [official code of DSR](https://github.com/qymeng94/DSR/blob/876c8d1ac41541b42d39cecdfce4d4b45bdcf365/cifar/modules/neuron.py#L170).
>
> - **B**: What is the definition of "true binary spikes"? Biologically, an action potential is triggered when the membrane potential is higher than a threshold. The output potential should NOT be a binary value in any case. The spike (action potential) is binary.``
>
>   - **Ans B**: During the spike operation the preceding layers will send either “0” or “1” activation/spike to the proceeding layers. This spike train reflects the ”true binary spike” which is either “0” or “1”. While in the DSR implementation, the output spikes are threshold dependent as mentioned in Eq. 7 of the original manuscript.
>
>     Consequently, the spikes to the proceeding layers, in this case, are not true binary [0,1] instead [0,$V_{th}$], where $V_{th}$ is learnable but also scaled by a hyperparameter $\alpha=0.3$.  In contrast, our proposed algorithm does not incorporate any kind of hyperparameter scalability and the generated spikes are “true binary” [0,1].
>
> - **C**: How hardware deals with layer-wise varied spikes is out of the scope of the paper. I assume the ``hardware" means neuromorphic hardware. Biologically, each neuron has a different dynamic threshold. The current neuromorphic hardware can barely mimic the simplest biological neuron. Thus, algorithm and theoretical study should not be limited or favored to existing neuromorphic hardware.
>
>   -  **Ans C**: Yes, this paper mainly deals with learnable potential threshold from the algorithm point of view. Regarding “hardware”, we are envisioning that SNN algorithms will eventually be deployed on some hardware (not limited to existing or any neuromorphic hardware), and no matter which hardware it is, we are just pointing out that true binary spikes of 0 or 1 will be much simpler to implement, compared to spikes that require layer-wise scaling parameters. In the last paragraph of Section 4 in the revised manuscript, we toned down the hardware related statements.
>
> - **D**: DSR and LTSNN used different base models,  VGG-11 for DSR and  VGG-9 for LTSNN. Is it fair?
>
>   - **Ans D**: Thank you for raising this concern. We performed additional experiments for LTSNN with the same VGG-11 architecture reported in DSR, for apple-to-apple comparison. The proposed LT-SNN improved the classification accuracy by 3.81% for DVS-CIFAR10 dataset as mentioned in the table below. We have added this table in the Appendix section of the revised manuscript as Table 7.
>
>     Besides using the same model as prior works, LT-SNN is validated with different models including compact ones, which helps us understand the Pareto curve of accuracy vs. model size as shown in Figure 1, where LT-SNN shows the SoTA results.
>
>    *[Table: Baseline comparison of proposed SGP with current SoTA.]*
>
> | **Method**    | **Architecture** | **Threshold** | **SG Func.** | **Top-1 Accuracy** |
> | ------------- | ---------------- | ------------- | ------------ | ------------------ |
> | TET[R1]       | VGG-11           | Fixed         | Triangle     | 77.33              |
> | DSR[R6]       | VGG-11           | Fixed         | STSG         | 75.70              |
> | **This work** | **VGG-11**       | **Learnable** | **SGP**      | **79.51**          |

---

> > ### Author Response · Authors · 2022-11-19
> > **Response to Reviewer Dob4: Q2 to Q3**
> >
> > **Q2**: The falsified hypothesis 1 is based only on  VGG-9 and DVS-CIFA10 is not convincing. In other words, how general of observation 1? In addition, I am not convinced by the experiments associated with Figure 2. If we stop training STSG before 80 epoch, STSG also has stabilized training, right? In other words, why the authors stopped training LTSNN at epoch 100? What would happen if the authors train LTSNN 200 epoch?
> >
> > **A2**: Following the reviewer’s suggestion, we performed additional experiments and trained both STSG and LT-SNN for 200 epochs. We use the same VGG-9 architecture as Figure 2 of the original manuscript and affirm that 200 epochs don’t impair the training stability of LT-SNN. In the revised manuscript, We updated Figure 2(b) with following 200 epochs training results, which demonstrates the stability of LT-SNN with extended training time.
> >
> > **Q3**: Super unclear of section 5.3! The L in Eq.12 is the same as the one in Eq.13? What's $L_{CE}$, MSE, $O(t)$, and T? More importantly, why the MSE regularizer is necessary?
> >
> > **A3**: We have incorporated output regularization to further improve the network performance.
> > $L$ in Eq. 12 represents the total loss (class loss + regularization loss) while  $L$ in Eq. 13 denotes the total number of layers. Thank you for pointing this out, we have changed the notations for ”loss” and ”number of layers” in Eq. 12 and Eq. 13 as presented in the equations below and also updated it in the revised manuscript.
> >
> > $L = \beta \frac{1}{T} \sum_{t=1}^{T}L_{CE}(O(t), y) + (1-\beta)\frac{1}{T}\sum_{t=1}^{T} MSE(O(t), \Omega)$
> >
> > Where $ \qquad\Omega = \frac{1}{\boldsymbol\ell} \sum_{l=1}^{\boldsymbol\ell}V^l_{th}$
> >
> >
> > Where $L$ represents the loss and $\boldsymbol\ell$ represents the total number of layers in the networks.
> > $L_{CE}$ iis the cross-entropy loss between the predicted and target class, MSE is the mean square error between the output spikes and average learnable potential threshold of all the layers, $O(t)$ represents the output spikes at the final layer and T represents the timestamps. MSE regularizer is incorporated to maximize the spike activity based on the optimized learnable threshold at the final layer. Eventually, it helps in class prediction at the fully connected layer. As described in the original manuscript, we adopt such cross-entropy-based temporally-averaged loss from [R1].
> > A concise version of the explanations above has been added to Section 5.3 to improve the clarity, and we also updated the title of Section 5.3.

---

> > > ### Author Response · Authors · 2022-11-19
> > > **Response to Reviewer Dob4: Q4**
> > >
> > > **Q4**: For the experiments associated with Table 4, why did the authors not use the same architecture of other competing approaches? What's the logic behind the experimental design?
> > >
> > > **A4**: Thank you for your question. Following the reviewer’s suggestion, we performed additional experiments with some same model architectures as prior works. As shown in the updated Table 4 of the revised manuscript, we added the additional experimental results of VGG-11, which has been employed in prior works.
> > > Besides using the same model as prior works, we verify the versatility of the proposed algorithm by applying LT-SNN to more compact model architectures such as 4-bit MobileNet-V1. This helps us understand the Pareto curve of accuracy vs. model size as shown in Figure 1, where LT-SNN shows the SoTA results.

---

> ### Comment · Reviewer_Dob4 · 2022-11-21
> **Post rebuttal comments**
>
> I appreciate the authors' efforts to clarify and answer my questions! It is not correct to falsify a hypothesis based on a specific model but testing with other models. The way to falsify a hypothesis itself needs to be clarified. I suggest writing in a way purely based on experimental results.
>
> The authors did not address my concerns about dynamic threshold. After checking other reviewers' comments and the rebuttal, I keep my current rate and do NOT suggest accepting the submission.

---

> > ### Author Response · Authors · 2022-12-10
> > **Response to Reviewer Dob4 Part 1**
> >
> > ```
> > I appreciate the authors' efforts to clarify and answer my questions! It is not correct to falsify a hypothesis based on a specific model but testing with other models. The way to falsify a hypothesis itself needs to be clarified. I suggest writing in a way purely based on experimental results.
> >
> > The authors did not address my concerns about dynamic threshold. After checking other reviewers' comments and the rebuttal, I keep my current rate and do NOT suggest accepting the submission.
> > ```
> >
> > Thank you for your additional comment. To further demonstrate the incompatbility and instability of the straight-through surrogate gradient (STSG) in Figure 2, we conducted additional experiment with STSG on MobileNet-V1 (Lite) model. As shown in the table below, the light-weight architecture reaches convergence with STSG with the cost of the degraded accuracy on the DVS-CIFAR10 dataset.
> >
> > | **Model**            | **SG Function** | **Top-1 Accuracy** |
> > |:--------------------:|:---------------:|:------------------:|
> > | MobileNet-V1 (Light) | STSG            | 72.10              |
> > | MobileNet-V1 (Light) | SGP             | 75.70              |
> >
> > We also performed extensive experiments to justify the fine-grained learning rate scaling is not beneficial to membrane potential optimization, as summarized in our responses to reviewer btJV.
> >
> > To answer your question falsifying hypothesis, we performed additional experiments on fine-grained learning rate tuning (lr) for weights ($W$) and $V_{th}$, with lr-$W$=0.001 and lr-$V_{th}$=0.0001, lr-$W$=0.001 and lr-$V_{th}$=0.00025, and lr-$W$=0.001 and lr-$V_{th}$=0.0005, and the results are reported in the table below.
> > Experiments with different learning rates could resolve the issue of abnormal training, but still at the cost of accuracy degradation as reported in the table below. We believe that what we attempted to point out regarding Hypothesis 1 is a general observation applicable to a number of experimental results that exhibit abnormal training or sub-optimal accuracy.
> >
> > Still, we agree that Hypothesis 1 and the way of falsifying it were not clear. To that end, to enhance clarity and avoid confusion, regarding 'Hypothesis 1' and 'Observation 1', in the next version of our manuscript, we will remove the hypothesis and observation, and instead describe the observed facts based purely on our original and newly added experimental results, as the reviewer suggested.
> >
> > *[Table: Performance comparison on DVS-CIFAR10 dataset with different learning rates and initial potential threshold values.]*
> >
> > | **Architecture** | **Lr-W** | **Lr-Vth** | **Initial Vth** | **SGP** | **Top-1 Accuracy (%)** |
> > |------------------|----------|------------|-----------------|---------|------------------------|
> > | VGG-9            | 0.1      | 0.001      | 1               | No      | 20.40                  |
> > | VGG-9            | 0.001    | 0.00001    | 1               | No      | 77.20                  |
> > | VGG-9            | 0.001    | 0.0001     | 1               | No      | 78.35                  |
> > | VGG-9            | 0.001    | 0.00025    | 1               | No      | 76.50                  |
> > | VGG-9            | 0.001    | 0.0005     | 1               | No      | 78.60                  |
> > | VGG-9            | 0.001    | 0.0005     | 2               | No      | 77.80                  |
> > | VGG-9            | 0.001    | 0.0005     | 4               | No      | 77.40                  |
> > | VGG-9            | 0.001    | 0.0005     | 6               | No      | 72.70                  |
> > | **VGG-9**        |**0.001** |**0.001**   | **1**           | **Yes** | **80.07**              |

---

> > > ### Author Response · Authors · 2022-12-10
> > > **Response to Reviewer Dob4 Part 2**
> > >
> > > ```
> > > In addition, many works have developed different dynamic threshold schemes for SNNs. The authors did not mention them at all. For example, in "Biologically Inspired Dynamic Thresholds for Spiking Neural Networks," the authors define dynamic thresholds without learning. More importantly, I do not think the proposed approach is able to imitate "the self-adaptiveness of the biological nervous system."
> > > ```
> > >
> > > We sincerely apologize that we did not respond to this specific comment in the first rebuttal, especially regarding the relevant paper that the reviewer pointed out. This paper [R3] %pointed out by the reviewer, presented a dynamic temporal threshold (DTT) method which introduces the layer-wise and neuron-wise dynamic potential threshold into spiking neural networks. However, our proposed LT-SNN is essentially different from the DTT algorithm [R3] in the following aspects
> > > - Our proposed LT-SNN algorithm considers the layer-wise potential threshold as learnable parameters which are optimized by the gradient-based training algorithm. On the contrary, DTT updates the neuron-wise thresholds based on the membrane potential difference between two consecutive timestamps.
> > > - DTT algorithm is only validated on the small-sized MNIST dataset with spiking CNN and LIF model, which has limited practicality with more complex scenarios and tasks (e.g., object detection with Prophesse Gen 1 dataset). On the contrary, the proposed LT-SNN algorithm is evaluated and benchmarked with various model architectures on multiple sizable datasets (CIFAR-10, DVS-CIFAR10, N-Caltech101, N-Cars, and Prophesse Gen 1).
> > >
> > > Furthermore, in biological nervous systems, it is true that each neuron has a different threshold value  [R1].
> > > On the algorithm level, associating every single receptive field (neurons) of SNN with a corresponding high-precision threshold value will largely increase the computation complexity and memory consumption of SNN, regardless of the computing platforms (e.g., GPU, FPGA, ASIC). Furthermore, the over-complicated AI models and expensive computation also does not align with the high energy efficiency of human brain (e.g., 20 Watt).
> > > Therefore, the proposed LT-SNN algorithm balances the biological complexity and the practical AI applications with layer-wise dynamic threshold design rather than the neuron-wise adaptive threshold.
> > >
> > > As we addressed the latest questions that were raised by the reviewer and all other reviewers along with the comments from John Willian, please kindly reconsider the overall evaluation of our paper.
> > >
> > > **References**
> > >
> > > [R1] S. Sardi et al. "New types of experiments reveal that a neuron functions as multiple independent threshold units." *Nature Scientific reports*, 2017.
> > >
> > > [R2] P. Etienne et al. "Learning to detect objects with a 1 megapixel event camera." *Advances in Neural Information Processing Systems (NeurIPS)*, 2020.
> > >
> > > [R3] J. Ding et al. "Biologically Inspired Dynamic Thresholds for Spiking Neural Networks." *Advances in Neural Information Processing Systems (NeurIPS)*, 2022.

---

### Official Review · Reviewer_kp4g · 2022-11-02

**Confidence:** 4
**Clarity, Quality, Novelty And Reproducibility:** The paper is written clearly and quit…
**Correctness:** 3
**Technical Novelty And Significance:** 3
**Empirical Novelty And Significance:** 3
**Recommendation:** 8

**Strength And Weaknesses:**


Strength
The paper offers a new view on learning adaptive thresholds in SNNs. Their SNNs acheve competitive performance on standard tasks even with by an order of magnitude fewer parameters. The paper is well organized and clearly written.

Weaknesses
While the authors bring up the question of biological relevance of SNNs and their learning mechanisms, they don't elaborate on this topic and don't provide evidence that their approach is more biologically realistic as compared to other SNN training schemes.
The authors do not comment if their code will be publicly available raising questions about replicability of their research.



**Summary Of The Paper:**

The authors address the thorny problem of training spiking deep neural networks that are marred with problems of non-differentiable nonlinearity, sparse responses and limited expressive power. The authors propose an adaptive threshold mechanism so that the threshold is trained using a different differentiation path from that used for the weight updates.
In extensive experiments the authors demonstrate advantages of their approach.

**Summary Of The Review:**

This is an impressive piece of work substantially improving on SOTA SNN training. However, questions remain about replicability of this work.

---

> ### Author Response · Authors · 2022-11-19
> **Response to Reviewer kp4g**
>
> **Q1**: While the authors bring up the question of biological relevance of SNNs and their learning mechanisms, they don't elaborate on this topic and don't provide evidence that their approach is more biologically realistic as compared to other SNN training schemes. The authors do not comment if their code will be publicly available raising questions about replicability of their research.
>
> **A1**: Thank you for raising this question.
> Thank you for raising this question. The main objective of this work is to improve the learnability and accuracy of SNNs. While we do get inspiration from biological nervous systems on dynamic neuronal thresholds, we focus on whether our specific proposed scheme can improve the accuracy compared to prior SNN works, not necessarily trying to make our approach more biologically realistic compared to other SNN works. By adopting the concept of the dynamic potential threshold in the nervous system [R8], the proposed LT-SNN achieves both high accuracies with more compact model sizes, as shown in Figure 1 of the original manuscript. We have made several modifications in Section 1 of the revise manuscript to clarify the biological inspiration and our objective of improving the SNN accuracy. We also removed a few occurrences of ambiguous biological plausibility.
>
> The code will be open-sourced for the research community, once the decision for our submission is finalized.

---

### Official Review · Reviewer_btjV · 2022-11-03

**Confidence:** 5
**Correctness:** 3
**Technical Novelty And Significance:** 2
**Empirical Novelty And Significance:** 2
**Recommendation:** 5

**Clarity, Quality, Novelty And Reproducibility:**

Clarity: clear, but should state the training setting more clearly in sec 5. Refer to weakness 1.

Quality: This work makes sense, but does not provide juicy information.

Novelty: The idea of training thresholds is not novel. But the authors make the idea work for the surrogate training framework using a simple trick.

Reproducibility: I think the method is easy to implement.

**Strength And Weaknesses:**

Strength:

1. The proposed method is simple, and can be directly plugged into other existing methods.

2. There are experiments on non-classification tasks. It is encouraged to explore the performance of SNNs on various tasks.

3. It is interesting to see the potential of quantized small-scale SNNs.

Weaknesses:

1. In table 2, does "GPW for all" mean that all neurons share the same learnable Vth? And does "SGP" mean that each layer has its own Vth? For the experiments shown in Fig. 2, does each layer have its own trainable Vth?
For each experiment, the authors need to make clear statements on whether all the neurons share the same Vth, or each layer has its own Vth, or each neuron has its own Vth. I also suggest the authors to combine eqns. 6 and 11 in one place, and clarify the range of the summation in eqn. 11.


2. I think the catastrophic performance drop in fig. 2a is not hard to understand. Let me first guess that each layer has its own trainable Vth. For one layer, you need to sum up dL/dVth_i for each neuron i in that layer to get dL/dVth. Since dL/dVth_i, dL/du_i, and dL/dWij have the same order, |dL/dVth| will be hundreds of or thousands of times large than |dL/dWij|, depending on the model width. Therefore, a learning rate suitable for updating Wij may be too large for updating Vth. So we can easily overcome the issue by using a small learning rate for Vth (e.g., set init lr=0.1 and 0.001 for W and Vth, respectively). Your proposed method basically shrinks dL/dVth_i with a sigmoid function, so it has a similar effect as using smaller lr.  Can you compare the two methods experimentally? Can you show the benefit of your proposed voltage-dependent shrinkage scheme?

3. I am concerned about the direct performance gain of the proposed method. The authors do not show the performance when disabling threshold training and keeping other settings the same. So it is not clear whether the sota results are achieved due to the proposed method or, say, network structures. Table 2 seems to show the direct effect of the proposed method, but the baseline uses a different network structure (the title of table 2 is misleading!). I would appreciate it if the authors could conduct ablation studies.

Minor:
1. Fig.1: DSR-VGG11 adopts learnable thresholds, as repeatedly pointed out in the manuscript.
2. In eqns. 6 and 9, a minus sign is missed. ds/du and ds/dVth have opposite signs.

**Summary Of The Paper:**

This paper proposes an effective way to train the spike thresholds of spiking neural networks, while the naive surrogate gradient method fails to do so. The proposed method achieves SOTA performance on several image classification tasks and an object detection task.

**Summary Of The Review:**

I give a "reject" because:

1. I challenge sec 4 and the motivation of the proposed method totally. You do not need to create a hypothesis and then give some observations. The issue is easy to handle: shrink dl/dVth or shrink the corresponding lr. See weakness 2.

2. The experiments are not carefully designed. I give some examples as following. In fig. 2(a), you do not fine-tune the lr for training vth (see bullet 1). In tables 1 and 2, DSR and TET use vgg-11, but you use vgg-9 (btw, I am confused about the term "true binary spike", DSR just uses binary spike). No ablation studies, which are super important in this work, see weakness 3.

---

> ### Author Response · Authors · 2022-11-19
> **Response to Reviewer bjtV: Major Q1**
>
> **Q1:** In table 2, does "GPW for all" mean that all neurons share the same learnable Vth? And does "SGP" mean that each layer has its own Vth? For the experiments shown in Fig. 2, does each layer have its own trainable Vth? For each experiment, the authors need to make clear statements on whether all the neurons share the same Vth, or each layer has its own Vth, or each neuron has its own Vth. I also suggest the authors to combine eqns. 6 and 11 in one place, and clarify the range of the summation in eqn. 11.
>
> **A1:** Thank you for your questions.
> Thank you for your questions. The major objective of the proposed LT-SNN algorithm is to achieve superior performance by smartly learning the membrane potential threshold during SNN training. In Section 4 of the original manuscript, we demonstrate the incompatibility between the widely-used straight-through surrogate gradient (STSG) function and threshold training, which motivated us to propose the Separate Gradient Path (SGP). SGP introduces the gradient penalty window (GPW), which is designed for gradient surrogation of threshold learning and leads to the stabilized training process, as shown in Figure. 2(b). Please note that both SGP and GPW are designed to optimize the layer-wise membrane potential threshold, i.e., each layer has a single optimized threshold value, and all channels of the layer are sharing the same threshold for spiking. GPW stabilizes the learning process of LT-SNN, which motivates us to investigate the following question: *Will the gradient penalty window (GPW) be suitable for all the gradient calculations?*
> To answer the above question, in the original manuscript we performed the experiment by applying the Sigmoid-based GPW to all the surrogate gradient computations of the entire  VGG-9 SNN model (both $\partial S/\partial u$ and $\partial S/\partial V_{th}$), abbreviated as "GPW for all''.  As summarized in Table 2 of the original manuscript, the degraded performance of “GPW for all” implies the necessity of the proposed Separated Gradient Path (SGP). In summary, both SGP and “GPW for all” optimizes SNN with layer-wise learnable threshold values. As shown in Figure 4(a) of the original manuscript, each convolution layer of VGG-9 will have a single layer-wise adaptive potential threshold value.
> Following the reviewer’s suggestion, we updated Eq. 11 with detailed gradient computation information. Since the learnable threshold is a single value, we reduce the dimensionality of the incoming gradient by summing the gradient along the batch, channel, height, and width. Based on your suggestions, we updated Eq. 11 accordingly in the revised manuscript.
>
> The reviewer is correct that Eq. 11 needs a clear range of the summation. We have included the range of summation in the updated Eq. 11 in the revised manuscript as follows:
> $$
> \begin{equation}
>    |\frac{\partial L}{\partial V_{th}}| = \frac{\partial L}{\partial S_t}\frac{\partial S_t}{\partial V_{th}} = \frac{\partial L}{\partial S_t} \sum_{(N,C,H,W)} \big(\mathbf{1}\{u_t\geq V_{th}\} \times \theta'(u_t-V_{th})\sigma(u_t-V_{th}))
> \end{equation}
> $$
> Here N, C, H, and W represent the batch size, the number of channels, frame height, and width, as we are summing over the pixel values of all the frames. Further, the gradient can be computed with the following PyTorch-like pseudocode:
>
> `dLdV = torch.sum(dLdV.mul(spike)).view(-1).mul(-1)`

---

> > ### Author Response · Authors · 2022-11-19
> > **Response to Reviewer bjtV: Major Q2**
> >
> > **Q2:** I think the catastrophic performance drop in fig. 2a is not hard to understand. Let me first guess that each layer has its own trainable Vth. For one layer, you need to sum up $dL/dVth_i$ for each neuron i in that layer to get $dL/dVth$. Since $dL/dVth_i$, $dL/du_i$, and $dL/dWij$ have the same order, $|dL/dVth|$ will be hundreds of or thousands of times large than $|dL/dWij|$, depending on the model width. Therefore, a learning rate suitable for updating Wij may be too large for updating Vth. So we can easily overcome the issue by using a small learning rate for Vth (e.g., set initial lr=0.1 and 0.001 for W and Vth, respectively). Your proposed method basically shrinks $dL/dVth_i$ with a sigmoid function, so it has a similar effect as using smaller lr. Can you compare the two methods experimentally? Can you show the benefit of your proposed voltage-dependent shrinkage scheme?
> >
> > **A2:** Thank you for this question with a very detailed explanation. To address this question, we performed additional experiments based on straight-through surrogate gradient (e.g., as we challenged in Hypothesis 1) with a lower learning rate on threshold training. The following table summarizes the additional experimental results with different learning rate schemes:
> >
> > *[Table1: Performance comparison of SGP and lower learning rate to balance out $V_{th}$ and $W_{ij}$ for LT-SNN training.]*
> >
> > | **Model** | **W lr**  | **Vth lr** | **SGP** | **Accuracy** | **Converged** |
> > | --------- | --------- | ---------- | ------- | ------------ | ------------- |
> > | VGG-9     | 0.1       | 0.001      | No      | 20.4         | No            |
> > | VGG-9     | 0.001     | 0.00001    | No      | 77.2         | Yes           |
> > | **VGG-9** | **0.001** | **0.001**  | **Yes** | **80.04**    | **Yes**       |
> >
> > The empirical results above indicate the sub-optimality of the separate learning rate scheduling. The proposed gradient penalty window (GPW) method not only penalizes the gradient magnitude of the learnable threshold, but also re-shapes the gradient surrogation during the backward propagation. SGP applies layer-wise dynamic gradient reshaping in the backward pass of training for optimal learnability, which is not possible with the fixed scaling of lr for $V_{th}$ with respect to the weight $W$.
> >
> > Furthermore, we compared the performance of various GPW functions (ArcTan, Triangle, Piecewise, and Sigmoid), and the experimental results below show that the Sigmoid GPW function has the optimal performance:
> >
> > *[Table: Comparison of different surrogate functions' performance as SGP.]*
> >
> > | **Model** | **Epochs** | **SGP**     | **Accuracy** | **Converged** |
> > | --------- | ---------- | ----------- | ------------ | ------------- |
> > | VGG-9     | 200        | ArcTan      | 77.81        | Yes           |
> > | VGG-9     | 200        | Triangle    | 70.83        | Yes           |
> > | VGG-9     | 200        | Piece-wise  | 78.81        | Yes           |
> > | **VGG-9** | **200**    | **Sigmoid** | **80.04**    | **Yes**       |
> >
> > We added the above tables in the Appendix section of the revised manuscript as Table 8 and Table 11.

---

> > > ### Author Response · Authors · 2022-11-19
> > > **Response to Reviewer bjtV: Major Q3**
> > >
> > > **Q3:** I am concerned about the direct performance gain of the proposed method. The authors do not show the performance when disabling threshold training and keeping other settings the same. So it is not clear whether the sota results are achieved due to the proposed method or, say, network structures. Table 2 seems to show the direct effect of the proposed method, but the baseline uses a different network structure (the title of Table 2 is misleading!). I would appreciate it if the authors could conduct ablation studies.
> > >
> > > **A3:** Thank you for pointing this out. We conducted additional ablation study experiments by disabling the learnable threshold while keeping the other settings to be identical. The following table summarizes the classification results on DVS-CIFAR10 dataset with VGG-9 and MobileNetV1 (Lite) architectures:
> > >
> > > *[Table: Performance comparison of learnable potential threshold and fixed potential threshold for the classification networks.]*
> > >
> > > | **Method**    | **Architecture**       | **Threshold** | **SG Func.** | **Top-1 Accuracy** |
> > > | ------------- | ---------------------- | ------------- | ------------ | ------------------ |
> > > | This work     | VGG-9                  | Fixed         | Triangle     | 78.02              |
> > > | **This work** | **VGG-9**              | **Learnable** | **SGP**      | **80.04**          |
> > > | This work     | MobileNet-v1(Lite)     | Fixed         | Triangle     | 74.70              |
> > > | **This work** | **MobileNet-v1(Lite)** | **Learnable** | **SGP**      | **75.70**          |
> > >
> > > In addition to the DVS-CIFAR10 dataset, we validated the performance of LT-SNN on Prophesee Gen1 dataset with Customized-YOLOV2-SNN:
> > >
> > > *[Table: Performance comparison of learnable potential threshold and fixed potential threshold for the object detection networks.]*
> > >
> > > | **Method**    | **Architecture**      | **Threshold** | **SG Func.** | **mAP**   |
> > > |---------------|-----------------------|---------------|--------------|-----------|
> > > | This work     | custom-Yolov2-SNN     | Fixed         | Triangle     | 0.122     |
> > > | **This work** | **custom-Yolov2-SNN** | **Learnable** | **SGP**      | **0.298** |
> > >
> > > Both of the above tables are added to the Appendix of the revised manuscript as Table 9 and Table 10. The empirical results in the above tables justify the superior performance of the proposed LT-SNN algorithm compared to the fixed potential threshold learning.
> > >
> > > Regarding Table 2, in the revised manuscript, we updated the first row of Table 2 with fixed threshold-based VGG-9 results (instead of VGG-11 results from TET), This way, now Table 2 of the revised manuscript shows the apple-to-apple comparison of the proposed SGP with other methods for the same VGG-9 model (and thus we kept the same title for Table 2).

---

> > > > ### Author Response · Authors · 2022-11-19
> > > > **Response to Reviewer bjtV: Minor Q1:Q2**
> > > >
> > > > **Q1**: Fig.1: DSR- VGG11 adopts learnable thresholds, as repeatedly pointed out in the manuscript.
> > > >
> > > > **A1**: Thank you for pointing this out. The reviewer is indeed correct, for DSR it should be "Learnable threshold" instead of fixed threshold. We have updated Figure 1 in the revised manuscript.
> > > >
> > > > **Q2**: In eqns. 6 and 9, a minus sign is missed. ds/du and ds/dVth have opposite signs.
> > > >
> > > > **A2**: In the original manuscript, we have used the absolute sign for Eq. 6 and Eq. 9. We have updated Eq. 6 and Eq. 9 in the revised manuscript (as below) with proper signs according to the reviewer's suggestion.
> > > >
> > > > $$
> > > > \begin{equation}
> > > >    \frac{\partial L}{\partial V_{th}} = -\frac{\partial L}{\partial S_t}\frac{\partial S_t}{\partial V_{th}} =
> > > >  -\frac{\partial L}{\partial S_t}\theta'(u_t-V_{th})
> > > > \end{equation}
> > > > $$
> > > >
> > > > $$
> > > > \begin{equation}
> > > >    \frac{\partial S_t}{\partial V_{th}} = -\theta'(u_t-V_{th}) \sigma(u_t-V_{th}) = -\max(0, 1-|u_t-V_{th}|)\mathbf{\sigma}(u_t-V_{th})
> > > > \end{equation}
> > > > $$

---

> > > > > ### Author Response · Authors · 2022-11-19
> > > > > **Response to Reviewer bjtV: Summary Q1:Q2**
> > > > >
> > > > > **Q1**: I challenge sec 4 and the motivation of the proposed method totally. You do not need to create a hypothesis and then give some observations. The issue is easy to handle: shrink dl/dVth or shrink the corresponding lr. See weakness 2.
> > > > >
> > > > > **A1**: As we detailed in our response for the reviewer’s comments on weakness 2, the additional experiments where we shrank the corresponding lr led to sub-optimal performance. SGP applies layer-wise dynamic gradient reshaping in the backward pass of training for optimal learnability, which is not possible with the fixed scaling of lr for $V_{th}$ with respect to the weight $W$.
> > > > >
> > > > > **Q2**: The experiments are not carefully designed. I give some examples as following. In fig. 2(a), you do not fine-tune the lr for training vth (see bullet 1). In tables 1 and 2, DSR and TET use  VGG-11, but you use  VGG-9 (btw, I am confused about the term "true binary spike", DSR just uses binary spike). No ablation studies, which are super important in this work, see weakness 3.
> > > > >
> > > > > **A2**: We have provided additional results for different learning rate values (added as Table 8 in the Appendix of the revised manuscript) and validated the superiority of our approach.Secondly, our main motivation is to achieve SoTA performance with a minimum number of parameters as mentioned in Figure 1. Therefore, we have demonstrated different light architectures in the paper including VGG-7, VGG-9, and MobileNet-light. However, following the reviewer’s recommendation, we have trained the VGG-11 architecture reported by DSR and TET using the proposed SGP with the layer-wise learnable threshold. The table below presents an apple-to-apple comparison of proposed LT-SNN with DSR and TET works. We have recorded a 3.81% and 2.18% accuracy improvement using LT-SNN, compared to the DSR and TET works respectively. We have included the table below in the Appendix section of the revised manuscript as Table 7.
> > > > >
> > > > > *[Table: Baseline comparison of proposed SGP with current SoTA.]*
> > > > >
> > > > > | **Method**    | **Architecture** | **Threshold** | **SG Func.** | **Top-1 Accuracy** |
> > > > > | ------------- | ---------------- | ------------- | ------------ | ------------------ |
> > > > > | TET[R1]       | VGG-11           | Fixed         | Triangle     | 77.33              |
> > > > > | DSR[R6]       | VGG-11           | Fixed         | STSG         | 75.7               |
> > > > > | **This work** | ** VGG-11**      | **Learnable** | **SGP**      | **79.51**          |

---

> > ### Comment · Reviewer_btjV · 2022-11-20
> > **Still have some concerns**
> >
> > Thanks for the reply. The authors solve some of my questions, but I still have several concerns:
> >
> > 1. The authors conduct experiments on training Vth with a smaller learning rate, but do not fine-tune the lr. I think the result of training Vth with a smaller lr could be improved for other Vth_lr (e.g., 0.0001 or 0.0005) and other initial Vth (e.g., 2,4,6).
> >
> > 2. I am not convinced that the gradient surrogation reshaping effect of GPW is important. The table [Table: Comparison of different surrogate functions' performance as SGP.] seems to show that different gradient reshaping has a great effect, but I think the main reason for the performance difference is the scaling effect of the SGs. For example, with commonly used hyperparameters, the max value of sigmoid SG is 0.25, while the value for actan SG is 0.5. BTW, the authors should give the formulae of the 4 SGs.
> >
> > 3. If the authors really aim at "achieving SoTA performance with a minimum number of parameters", they should compare their models with the real SoTA, especially for CIFAR-10. They also need to conduct experiments on ImageNet, which is a more complicated dataset.
> >
> > 4. The ablation studies are only conducted on dvscifar10 and Prophesee Gen1, but I am still worried about CIFAR-10. As pointed out in Unfair 1 of John Willian's comments, the proposed method leads to a performance drop on CIFAR-10. Then I also cannot evaluate the effectiveness of the proposed method for larger-scale datasets like cifar-100 and ImageNet. I would like to thank John Willian for his comments.
> >
> > The authors do not give me a chance to discuss with them. Based on their current rebuttal, I keep my score.

---

> > > ### Author Response · Authors · 2022-12-10
> > > **Response to Reviewer bjtV Part 1**
> > >
> > > ```
> > > **Conncern 1:** The authors conduct experiments on training Vth with a smaller learning rate but do not fine-tune the lr. I think the result of training Vth with a smaller lr could be improved for other $V_{th_lr}$ (e.g., 0.0001 or 0.0005) and other initial Vth (e.g., 2,4,6).
> > > ```
> > >
> > > Thank you for your additional comments. %raising this concern.
> > > We followed the reviewer's suggestion and conducted additional experiments with different initial $V_{th}$ values and learning rate combinations.
> > > In the first experiment, the learning rate of the learnable potential threshold $V_{th}$ and weights (W) are set to 0.0005 and 0.001 respectively. We swept over the initial membrane potential threshold by setting its initial values to 1.0, 2.0, 4.0, and 6.0. As shown in the table below, the fine-grained parameter tuning on the initial $V_{th}$ value failed to outperform the proposed SGP training scheme. We also performed additional 3 experiments with the proposed LT-SNN method and report the accuracy statistics below, where the consistently high accuracy consolidates the necessity of the proposed separate gradient path (SGP).
> > >
> > > *[Table: Performance comparison on DVS-CIFAR10 dataset with different learning rates and initial potential threshold values.]*
> > >
> > > | **Architecture** | **Lr-W** | **Lr-Vth** | **Initial Vth** | **SGP** | **Top-1 Accuracy (%)** |
> > > |------------------|----------|------------|-----------------|---------|------------------------|
> > > | VGG-9            | 0.1      | 0.001      | 1               | No      | 20.40                  |
> > > | VGG-9            | 0.001    | 0.00001    | 1               | No      | 77.20                  |
> > > | VGG-9            | 0.001    | 0.0001     | 1               | No      | 78.35                  |
> > > | VGG-9            | 0.001    | 0.00025    | 1               | No      | 76.50                  |
> > > | VGG-9            | 0.001    | 0.0005     | 1               | No      | 78.60                  |
> > > | VGG-9            | 0.001    | 0.0005     | 2               | No      | 77.80                  |
> > > | VGG-9            | 0.001    | 0.0005     | 4               | No      | 77.40                  |
> > > | VGG-9            | 0.001    | 0.0005     | 6               | No      | 72.70                  |
> > > | **VGG-9**        |**0.001** |**0.001**   | **1**           | **Yes** | **80.07**              |

---

> > > > ### Author Response · Authors · 2022-12-10
> > > > **Response to Reviewer bjtV Part 2**
> > > >
> > > > ```
> > > > **Conncern 2:** I am not convinced that the gradient surrogation reshaping effect of GPW is important. The table [Table: Comparison of different surrogate functions' performance as SGP.] seems to show that different gradient reshaping has a great effect, but I think the main reason for the performance difference is the scaling effect of the SGs. For example, with commonly used hyperparameters, the max value of sigmoid SG is 0.25, while the value for actan SG is 0.5. BTW, the authors should give the formulae of the 4 SGs.
> > > > ```
> > > >
> > > > Thank you for raising your concern on this. To address your question, we performed additional experiments by scaling down the learning rate of $V_{th}$ with different reduction factors. We agree with you that the max magnitude of the Sigmoid gradient penalty window (GPW) is 0.25. However, as shown in the following table with additional experimental results, scaling down the learning rate of $V_{th}$ by 0.25$\times$ (i.e., lr-$V_{th}$=0.00025) leads to sub-optimal accuracy with 3.57% lower accuracy, compared to the proposed method. On the contrary, a more gentle scaling factor such as 0.5 (i.e., lr-$V_{th}$=0.0005) leads to higher accuracy compared to the $0.25\times$ case. but the accuracy is still worse (1.47% lower) than the proposed separate gradient path (SGP) method. The experimental results show that \textbf{the learning process of threshold is not solely dependent on the learning rate}, and the proposed GPW and SGP methods optimize $V_{th}$ with the dedicated gradient computation landscape, overall leading to the improved accuracy.
> > > >
> > > > We appreciate the comment on the formuale of the 4 SGs. In the equations below, we have provided the 4 SGs of Arctan, Triangle, Piece-wise, and Sigmoid.
> > > >
> > > > \begin{align}
> > > >    \text{ArcTan}: f(u_t-V_{th}) = \text{Atan}(u_t-V_{th}) \\
> > > > \end{align}
> > > > \begin{align}
> > > >    \text{Triangle}: f(u_t-V_{th}) = \text{max}(0, 1-|u_t-V_{th}|) \\
> > > > \end{align}
> > > > \begin{align}
> > > >    \text{Piece-wise [R1]}: f(u_t-V_{th}) = -V_{th}^2|(u_t-V_{th})| + V_{th} \\
> > > > \end{align}
> > > > \begin{align}
> > > >    \text{Sigmoid}: f(u_t-V_{th}) = \sigma(u_t-V_{th}) = \frac{1}{1+e^{-(u_t-V_{th})}}
> > > > \end{align}
> > > >
> > > > *[Table:  Performance comparison on DVS-CIFAR10 dataset with differently scaled learning rate of potential threshold Vth.]*
> > > >
> > > > | **Architecture** | **Lr-W** | **Lr-Vth** | **Initial Vth** | **SGP** | **Top-1 Accuracy (%)** |
> > > > |------------------|----------|------------|-----------------|---------|------------------------|
> > > > | VGG-9            | 0.001    | 0.0001     | 1               | No      | 78.35                  |
> > > > | VGG-9            | 0.001    | 0.00025    | 1               | No      | 76.5                   |
> > > > | VGG-9            | 0.001    | 0.0005     | 1               | No      | 78.6                   |
> > > > | **VGG-9**        |**0.001** |**0.001**   |**1**            |**Yes**  |**80.07**               |

---

> > > > > ### Author Response · Authors · 2022-12-10
> > > > > **Response to Reviewer bjtV Part 3**
> > > > >
> > > > > ```
> > > > > **Conncern 3:** If the authors really aim at "achieving SoTA performance with a minimum number of parameters", they should compare their models with the real SoTA, especially for CIFAR-10. They also need to conduct experiments on ImageNet, which is a more complicated dataset.
> > > > > ```
> > > > >
> > > > > We believe that SNNs will be more suitable for event-based sensors (e.g. dynamic vision sensors) and corresponding neuromorphic datasets, because such sensors output binary spikes that represent events, and these could be directly connected to the inputs of the SNN models.
> > > > > On the other hand, for static image based datasets such as CIFAR-10 or ImageNet, the static images need to be converted into spike format, which incurs latency/power overhead, and this could diminish the benefits of SNNs but such overhead is not well accounted for in SNN works for CIFAR-10/ImageNet datasets (such overhead does not exist for conventional deep neural networks for CIFAR-10/ImageNet datasets).
> > > > > The spatial-temporal information captured by DVS sensors naturally fits the computation procedure of spiking neural networks, which is the major focus of the proposed LT-SNN algorithm. Also, please note that we validated the proposed LT-SNN algorithm on the large-sized Prophesee Gen1 dataset, which contains 228,123 bounding boxes for cars and 27,658 for pedestrians under the practical urban environment. The SoTA accuracy achieved by the proposed LT-SNN algorithm demonstrates the powerful performance on the large and practical event-based dataset.
> > > > >
> > > > > Still, based on the reviewer's feedback (and also based on John Willian's feedback), we reported the CIFAR-10 accuracy results using the identical setup with the open-sourced codes of TET [R2] and our LT-SNN shows superior performance for CIFAR-10 dataset.
> > > > > Compared to Dspike [R7], LT-SNN's accuracy for CIFAR-10 is 0.28% lower. The accuracy comparison to SoTA works for CIFAR-10 dataset is summarized in the following table.
> > > > > ImageNet experiment takes considerably more time, so we do not have the ImageNet accuracy results yet, but we will perform the ImageNet experiments as well.
> > > > > We will include the additional CIFAR-10 results (reported above) and ImageNet results (will obtain soon) in the next version of our paper.
> > > > >
> > > > > Regarding the reviewer's main concern here on "achieving SoTA performance", we will clarify in the next version of our paper that we achieve SoTA performance on "event-based" or neuromorphic datasets.
> > > > > We will clarify that Dspike [R7] has slightly higher accuracy on (non-event-based) CIFAR-10 dataset than ours, and we will see how the ImageNet accuracy results turn out compared to other works.

---

> > > > > > ### Author Response · Authors · 2022-12-10
> > > > > > **Response to Reviewer bjtV Part 4**
> > > > > >
> > > > > > ```
> > > > > > **Conncern 4:** The ablation studies are only conducted on dvscifar10 and Prophesee Gen1, but I am still worried about CIFAR-10. As pointed out in Unfair 1 of John Willian's comments, the proposed method leads to a performance drop on CIFAR-10. Then I also cannot evaluate the effectiveness of the proposed method for larger-scale datasets like cifar-100 and ImageNet. I would like to thank John Willian for his comments.
> > > > > > ```
> > > > > >
> > > > > > Thank you for your additional comments, and please take a look at our responses to John's comments.
> > > > > > Also, please note our emphasis on SNNs for DVS-type event-based sensors and corresponding event-based datasets, as we described in our response (A3) to your previous concern (Concern 3).
> > > > > > Still, we understand your concern on less amount of evaluations/comparisons reported in our original manuscript for CIFAR-10/100 and ImageNet datasets, so we detail our response on that aspect below.
> > > > > >
> > > > > > Our original manuscript did not report the CIFAR-10 results of the TET algorithm due to the non-reproducible accuracy reported in [R2]. As described in the Appendix A.1 of the [R2], TET is firstly trained by 300 epochs with 2 simulation steps (T=2), 0.01 learning rate on Adam optimizer, 512 batch size, and cosine annealing learning rate scheduler.
> > > > > > The authors of [R2] open-sourced their codes, so using the exact same codes from the open-sourced official implementation \href{https://github.com/Gus-Lab/temporal_efficient_training}{(official code)}, we ran the CIFAR-10 experiments under the same simulation length (T=2).
> > > > > > However, with this open-sourced official code for T=2, the best accuracy we could achieve on ResNet-19 model for CIFAR-10 dataset was 92.53%, which is 1.63% less than the reported accuracy (94.16%) in [R2].
> > > > > > In other words, $>$94% accuracy reported in [R2] was not reproducible with the authors' open-sourced codes.
> > > > > > Now, with the same experimental setup as the open-sourced code of TET, our proposed LT-SNN achieves 92.87%, which outperforms the accuracy achieved by the open-sourced codes of TET by 0.45%, as summarized in the following table:
> > > > > >
> > > > > > *[Table: Performance comparison of TET and LT-SNN on CIFAR-10 dataset with simulation length T=2.]*
> > > > > >
> > > > > > | **Method** | **Architecture** | **Threshold** | **SG Func** | **Top-1 Accuracy (%)** |
> > > > > > |------------|------------------|---------------|-------------|------------------------|
> > > > > > | TET[R2]    | ResNet-19        | Fixed         | Triangle    | 94.16 (Reported)       |
> > > > > > | TET[R2]    | ResNet-19        | Fixed         | Triangle    | 92.53 (Reproduced)     |
> > > > > > | **VGG-9**  | **ResNet-19**    | **Learnable** | **SGP**     | **92.7**               |
> > > > > >
> > > > > > Since the accuracy reported in the TET paper [R2] was not reproducible, we did not report the CIFAR-10 accuracy results in our original manuscript. We performed the CIFAR-10 experiment with $1e$-3 learning rate and batch size = 128. By training our LT-SNN for ResNet-19 with 200 epochs under simulation length of T=6, we achieved 93.97% inference accuracy.
> > > > > >
> > > > > > The non-reproducibility of the TET algorithm also occurred in the CIFAR-100 experiments that we additionally performed. With the same experimental setup and the official code of the paper, we can only achieve 72.07% CIFAR-100 accuracy with T=2, where the 0.8% accuracy difference is non-negligible.
> > > > > >
> > > > > > *[Table: Performance comparison of TET and LT-SNN on CIFAR-100 dataset with simulation length T=2.]*
> > > > > >
> > > > > > | **Method** | **Architecture** | **Threshold** | **SG Func** | **Top-1 Accuracy (%)** |
> > > > > > |------------|------------------|---------------|-------------|------------------------|
> > > > > > | TET[R2]    | ResNet-19        | Fixed         | Triangle    | 72.87 (Reported)       |
> > > > > > | TET[R2]    | ResNet-19        | Fixed         | Triangle    | 72.07 (Reproduced)     |
> > > > > > | **VGG-9**  | **ResNet-19**    | **Learnable** | **SGP**     | **72.62**              |
> > > > > >
> > > > > > We agree with the reviewer that evaluating the performance of SNN on conventional computer vision datasets (e.g., CIFAR-10/100) is still important. DSR [R6] reported the $>$95% CIFAR-10 accuracy on ResNet-18 with 20 simulation steps. The extensive simulation length leads to comparable accuracy as the full precision non-spiking DNN training.

---

> > > > > > > ### Author Response · Authors · 2022-12-10
> > > > > > > **Response to Reviewer bjtV Part 5**
> > > > > > >
> > > > > > > With the normal neural network training, the memory cost includes the weight memory and activation memory. On the contrary, the memory cost of SNN includes weight memory, membrane potential memory, and spike memory [R3]. While binary spikes are computationally efficient, holding the intermediate high-precision membrane potential for an extensive simulation length (e.g., T=20) is memory intensive.
> > > > > > > In addition to the memory cost, the iterative accumulation process of SNN also magnifies the energy consumption of training. With simulation length T=8, [R4] shows that the SNN training consumes 1.35$\times$ more energy compared to normal CNN training with VGG5 architecture on the CIFAR-10 dataset.
> > > > > > > Furthermore, the extensive simulation length only leads to marginal accuracy improvements (95.40% vs. 95.01%) in DSR [R6]. Given the observations from the recent works, we would like to point out the low practicality of the extensive time steps in both training and inference.
> > > > > > >
> > > > > > > Motivated by that, all of the CIFAR-10 results of the proposed LT-SNN algorithm are evaluated based on the low-simulation length (e.g., T$\leq$10 or less), which is also employed in the DSpike algorithm (pointed out by John Willian). The following table summarizes the performance of different SNN training algorithms with limited simulation length on the CIFAR-10 dataset:
> > > > > > >
> > > > > > > *[Table: Performance comparison SNN with limited simulation length on CIFAR-10.]*
> > > > > > >
> > > > > > > | **Method** | **Model** | **Simulation Length** | **Accuracy** | **Parameters** |
> > > > > > > |------------|-----------|-----------------------|--------------|----------------|
> > > > > > > | Hybrid[R8] | ResNet-18 | 10                    | 92.54        | 12.3M          |
> > > > > > > | tdBN[R5]   | ResNet-18 | 6                     | 93.16        | 11.17M         |
> > > > > > > | DSR[R6]    | ResNet-18 | 6                     | 91.89        | 11.17M         |
> > > > > > > | DSpike[R7] | ResNet-19 | 6                     | 94.25        | 11.18M         |
> > > > > > > | This work  | ResNet-19 | 6                     | 93.97        | 11.18M         |
> > > > > > >
> > > > > > > With a limited simulation length, the proposed LT-SNN algorithm achieves higher CIFAR-10 accuracy than most works from the SoTA literature. Only Dspike [R7] exhibits higher CIFAR-10 accuracy than ours, by a 0.28% margin.
> > > > > > > Note that [R7] proposed a surrogate gradient estimator, which can cover a large range of choices for surrogate gradient (SG), with the cost of introducing extra computation in the forward pass. As shown in Figure 3 of [R7], the accumulated membrane potential $u_t$ has to be windowed by the Dspike function to formulate output $y_t$. In the meantime, the trained temperature coefficient $b$ of the DSpike function converges to a non-significant value (Figure 5 of [R7]), which leads to the non-linear shape DSpike function (Figure 3, left, of [R7]).
> > > > > > > In practice, supporting such a highly-nonlinear function is cumbersome (especially considering hardware mapping), and performing the additional non-linear operation at every single time step is expensive during the forward pass, which makes the DSpike [R7] less efficient and hardware-compatible compared to our proposed LT-SNN algorithm.
> > > > > > > Admittedly our proposed LT-SNN has marginally less CIFAR-10 accuracy compared to DSpike [R7] (93.97% vs. 94.25%), but we believe the proposed LT-SNN achieves the best tradeoff between the hardware simplicity and algorithm-level accuracy.
> > > > > > >
> > > > > > > Please understand that we addressed all the questions and concerns raised by you as well as John Willian, and kindly reconsider your decision regarding our submission.
> > > > > > >
> > > > > > > **References**
> > > > > > >
> > > > > > > [R1] F, Wei, et al. "SpikingJelly", https://github.com/fangwei123456/spikingjelly
> > > > > > >
> > > > > > > [R2] S. Deng et al. "Temporal Efficient Training of Spiking Neural Network via Gradient Re-weighting." International Conference on Learning Representation (ICLR), 2022.
> > > > > > >
> > > > > > > [R3] D, Lei, et al. "Rethinking the performance comparison between SNNS and ANNS." Neural networks, 2020.
> > > > > > >
> > > > > > > [R4] R. Yin, et al. "SATA: Sparsity-Aware Training Accelerator for Spiking Neural Networks," in IEEE Transactions on Computer-Aided Design of Integrated Circuits and Systems, 2022.
> > > > > > >
> > > > > > > [R5] H. Zheng et al. "Going deeper with directly-trained larger spiking neural networks". AAAI Conference on Artificial Intelligence, 2021.
> > > > > > >
> > > > > > > [R6] Q. Meng et al. "Training High-Performance Low-Latency Spiking Neural Networks by Differentiation on Spike Representation." IEEE/CVF Conference on Computer Vision and Pattern Recognition (CVPR), 2022.
> > > > > > >
> > > > > > > [R7] Y. Li et al. "Differentiable spike: Rethinking gradient-descent for training spiking neural networks." Advances in Neural Information Processing Systems (NeurIPS), 2021.
> > > > > > >
> > > > > > > [R8] N. Rathi and K. Roy. "Diet-snn: Direct input encoding with leakage and threshold optimization in deep spiking neural networks." International Conference on Learning Representation (ICLR), 2021.

---

> ### Comment · Reviewer_btjV · 2022-12-11
> **My opinion based on the new rebuttal**
>
> The updated rebuttal addresses some of my concerns.
>
> 1. Now, I accept that SGP is smarter than vanilla learning rate scaling. In the next version of the manuscript, the authors can give some intuitive explanations about why such a voltage-dependent scaling technique works better.
>
> 2. Based on the additional ablation studies, I think SGP is effective, at least for moderate tasks. Two minor issues: (i) I don't know why the reported accuracy of TET cannot be reproduced by their official code. According to my self-implementation (I did not use the official code), TET can achieve 94%+ accuracy on CIFAR-10. I suggest the authors tune the hyperparameters to get better performance. (ii) DSR can also achieve 94%+ accuracy with T=5 on CIFAR-10, as reported in their paper.
>
> 3. I am glad that the authors will abandon the "Hypothesis 1 & Observation 1" story in their next version of the manuscript. The catastrophic performance drop is not a common issue. For example, in a recent paper [1], the proposed method doesn't seem to suffer from the issue. You only need to explain why SGP works better than vanilla threshold training.
>
> 4. I agree with the authors that "SNNs will be more suitable for event-based sensors and corresponding neuromorphic datasets". But the embarrassing thing is that there are no large-scale complicated neuromorphic datasets. DVS-CIFAR10 only contains 10000 samples in total, and is essentially the Poisson encoding of CIFAR-10. So we can also easily achieve good performance with an ANN (the ANN performance can be even better than the SNN performance). Therefore, to verify an SNN training method, we still have to conduct experiments on large-scale static datasets like ImageNet, as done by all recent SOTA.
>
> I would like to increase my score to 5 given the authors' great efforts in the rebuttal. I still do not give an "accept" since the next version is not finished: the ImageNet result and the new story are not shown.
>
>
> [1] Wang et al., LTMD: Learning Improvement of Spiking Neural Networks with Learnable Thresholding Neurons and Moderate Dropout.

---

### Public Comment · ~John_Willian1 · 2022-11-05
**Interesting work to follow! But, it seems to have some serious issues in the paper as well.**

With the proposed adaptive learning method, this work advanced the performance of SNNs on the event dataset CIFAR10-DVS with smaller neural network backbones, which is impressive!

However, I have been reading recent works relevant to SNNs, including all the latest submissions for ICLR 2023, and I noticed that, in this paper,

(Unfair 1)

the prior works listed in Row CIFAR10, Table 4 are incomplete. For example, TET, listed in Row DVS-CIFAR-10, Table 4 and **adopted** as a trick in the proposed LT-SNN, also conducts experiments on CIFAR-10 with ResNet-19 and Simulation Length 6 but is **not** listed in Row CIFAR10. And, their TET ResNet-19 with Simulation Length 6 can achieve 94.50 $\pm$ 0.07, which is **better** than LT- ResNet-19. **Since the proposed LT-SNN also adopts TET, the proposed method seems to impair the performance and is incompatible with TET on CIFAR10.**

(Unfair 2)

And, Dspike [1], proposed at Neurips 2021 and also with respect to **surrogate gradients**, is dedicated to improving the direct training of SNNs which I think is **very related** to this work. And Dspike also conducts experiments on both CIFAR10 and CIFAR10DVS, but is **neither** listed as a comparison **nor** cited by this paper.  **There should be a fair comparison between Dspike and LT-SNN (without TET)**


(Unfair 3)

As mentioned above, TET is adopted in the proposed method and the TET baseline but is not adopted in other baseline methods. **It is unfair that the authors use an advanced trick TET to beat other works that don't.**


(Unfair 4)

I also noticed the model architecture used in Table 4 is different from prior works.  It makes sense that the authors want to show the proposed method can help smaller models do better on CIFAR10DVS as claimed in Figure 1. However, to my best knowledge and experience, **1)** **CIFAR10DVS is a small dataset**, and the obstacle is over-fitting. So, that smaller models can do better than bigger models is not surprising, and thus does not back up the proposed method effectively.  **2)** I think it is very necessary to use **the same model architecture and hyper-parameters** in the main experiments (Table 4).  Just keep any other hyper-parameters consistent with the prior works.  It should be common sense to make fair comparisons when proving the proposed method is better than others. The authors have done so many experiments in Table 4. Why did they lose this vital one?


(Fairness Conclusion)

Is it because they are irrelevant to this work in the authors' opinion, due to an incomplete survey, or because the proposed method **failed to surpass them on CIFAR10**? Anyway, I think this work could be more convincing if sufficient and thorough comparisons are listed. Or, at least plugging the proposed method into those aforementioned works to demonstrate the effectiveness and superiority of the proposed method.  For now, I don't think the main experiments are convincing enough, and some doubts arise.

(Doubt 1)

Furthermore, DSR is proposed to enable direct-trained SNNs to have more simulation steps and meanwhile less training overhead (which is equal to only one simulation step in the backward pass).  Could the proposed method achieve comparable training efficiency? If not, I think treating DSR as the main baseline could be somewhat unfair.

(Doubt 2)

Figure 2(a) seems **not** to match what I have observed in my experiments which show **STSG performs well on the learnable potential threshold**. I tried to reproduce Figure 2(a), but never find the **Failed Training**. It seems like **Observation 1** is **not generous** in most scenarios.  The authors' falsification of **Hypothesis 1** is not persuasive since Figure 2(a) can hardly be observed. Thus, the motivation of the paper is not as concrete and solid as it claims.  Will they **open-source** their code to reproduce Figure 2(a) or the training logs? and when? I think this is very important.  Because it will be less meaningful if the whole work is based on minor special cases rather than generous circumstances.

I am a Ph.D. student in the SNN research field. My concerns are of course less important since I am not an official reviewer. But I really hope the authors could address my problems, which I think are very crucial!

And, forgive me for my harsh words, I am not intended to be aggressive. Noticing that two reviewers gave a rating of 8, which is impressive and attractive, made me decide to pay more attention to this paper to learn something valuable. It naturally raises some issues after my reading.

Again, this work is still, to my concept, innovative and progressive to some points. But, the experiments and the observations are very rough and unconvincing.

Thanks for the authors' time. I am very much looking forward to the reply.

[1] Li Y, Guo Y, Zhang S, et al. Differentiable spike: Rethinking gradient-descent for training spiking neural networks[J]. Advances in Neural Information Processing Systems, 2021, 34: 23426-23439.

---

> ### Author Response · Authors · 2022-12-10
> **Response to John Willian part 1**
>
> ```
> **Unfair 1.** the prior works listed in Row CIFAR10, Table 4 are incomplete. For example, TET, listed in Row DVS-CIFAR-10, Table 4 and adopted as a trick in the proposed LT-SNN, also conducts experiments on CIFAR-10 with ResNet-19 and Simulation Length 6 but is not listed in Row CIFAR10. And, their TET ResNet-19 with Simulation Length 6 can achieve 94.50$\pm$0.07, which is better than LT- ResNet-19. Since the proposed LT-SNN also adopts TET, the proposed method seems to impair performance and is incompatible with TET on CIFAR10.
> ```
>
> Thank you for raising this concern. Our original manuscript did not report the CIFAR-10 results of the TET algorithm due to the non-reproducible accuracy reported in [RJ1]. As described in the Appendix A.1 of the [RJ1], TET is firstly trained by 300 epochs with 2 simulation steps (T=2), 0.01 learning rate on Adam optimizer, 512 batch size, and cosine annealing learning rate scheduler.
> The authors of [RJ1] open-sourced their codes, so using the exact same codes from the open-sourced official implementation \href{https://github.com/Gus-Lab/temporal_efficient_training}{(official code)}, we ran the CIFAR-10 experiments under the same simulation length (T=2).
> However, with this open-sourced official code for T=2, the best accuracy we could achieve on ResNet-19 model for CIFAR-10 dataset was 92.53%, which is 1.63% less than the reported accuracy (94.16%) in [RJ1].
> In other words, $>$94% accuracy reported in [RJ1] was not reproducible with the authors' open-sourced codes.
> Now, with the same experimental setup as the open-sourced code of TET, our proposed LT-SNN achieves 92.87%, which outperforms the accuracy achieved by the open-sourced codes of TET by 0.45%. Furthermore, we performed the CIFAR-10 experiment for ResNet-19 with 200 epochs under the simulation length of T=6. We used $1e$-3 learning rate and batch size = 128. By training our proposed LT-SNN we achieved 93.97% inference accuracy which is 0.89% higher than the reproduced 93.09% accuracy, as summarized in the following table:
>
> *[Table: Performance comparison of TET and LT-SNN on CIFAR-10 dataset.]*
>
> | **Method** | **Architecture** | **T** | **Threshold** | **SG Function** | **Top-1 Accuracy**                 |
> |:----------:|:----------------:|:-----:|:-------------:|:---------------:|:----------------------------------:|
> | TET[RJ1]   | ResNet-19        | 2     | Fixed         | Triangle        | 94.16 (Reported, not reproducible) |
> | TET[RJ1]   | ResNet-19        | 2     | Fixed         | Triangle        | 92.53 (Reproduced)                 |
> |**This work** |**ResNet-19**   | **2** |**Learnable**  |**SGP**          |**92.87**                           |
> | TET[RJ1]   | ResNet-19        | 6     | Fixed         | Triangle        | 94.50 (Reported, not reproducible) |
> | TET[RJ1]   | ResNet-19        | 6     | Fixed         | Triangle        | 93.09 (Reproduced)                 |
> |**This work** |**ResNet-19**   |**6**  |**Learnable**  |**SGP**          |**93.97**                           |
>
> Since the accuracy reported in the TET paper [RJ1] was not reproducible, we did not report the CIFAR-10 accuracy results from the original TET paper in our original manuscript.
> In the next version of our paper, we will add the description and these results from the table above on CIFAR-10 accuracy and comparison in the main text and in the Appendix.

---

> > ### Author Response · Authors · 2022-12-10
> > **Response to John Willian part 2**
> >
> > ```
> > **Unfair 2.** And, Dspike [1], proposed at Neurips 2021 and also with respect to surrogate gradients, is dedicated to improving the direct training of SNNs which I think is very related to this work. And Dspike also conducts experiments on both CIFAR10 and CIFAR10DVS, but is neither listed as a comparison nor cited by this paper. There should be a fair comparison between Dspike and LT-SNN (without TET)
> > ```
> >
> > Thank you for pointing this paper to us. First of all, Dspike has reported 75.4±0.05% accuracy with ResNet-18 for the DVS-CIFAR-10 dataset, while the proposed LT-SNN has achieved 80.04%, which is 4.67% higher with 1.57$\times$ smaller model (11.17 Millon parameters vs. 7.07 Millon). To eliminate the performance variation caused by the different trials, we conducted 3 runs on the DVS-CIFAR10 dataset and the VGG-9 architectures, and the accuracy results are reported in the following table:
> >
> > *[Table: Performance comparison of TET and LT-SNN on DVS-CIFAR-10 dataset.]*
> >
> > | **Method**   | **Architecture** | **Param** | **SG Function** | **Top-1 Accuracy** |
> > |:------------:|:----------------:|:---------:|:---------------:|:------------------:|
> > | This work    | VGG-9            | 7.07M     | SGP             | 80.04              |
> > | This work    | VGG-9            | 7.07M     | SGP             | 79.97              |
> > | This work    | VGG-9            | 7.07M     | SGP             | 80.20              |
> > |**This work** |**VGG-9**         |**7.07M**  |**SGP**          |**80.07**           |
> > | DSpike [RJ5] | ResNet-19        | 11.17M    | DSpike          | 75.4               |
> >
> > Secondly, DSpike [RJ5] proposed a surrogate gradient estimator, which can cover a large range of choices for surrogate gradient (SG), with the cost of introducing extra computation in the forward pass. As shown in Figure 3 of [RJ5], the accumulated membrane potential $u_t$ has to be windowed by the Dspike function to formulate output $y_t$. In the meantime, the trained temperature coefficient $b$ of the DSpike function converges to a non-significant value (e.g., Figure 5 of [RJ5]) for the DVS-CIFAR10 dataset, which leads to the non-linear shape DSpike function (Figure 3, left, of [RJ5]). In practice, supporting such a highly-nonlinear function is cumbersome (especially considering hardware mapping), and performing the additional non-linear operation at every single time step is expensive during the forward pass, which makes the DSpike [RJ5] less efficient and hardware-compatible compared to our proposed LT-SNN algorithm. Admittedly our proposed LT-SNN has marginally less CIFAR-10 accuracy compared to the DSpike [RJ5] with ResNet models (93.97% vs. 94.25%), but we believe the high hardware compatibility of the proposed LT-SNN is more competitive than DSpike [RJ5].
> >
> > Regarding the fair comparison, we believe the most fair comparison is simply based on the inference accuracy.
> > TET might be thought of as an advanced technique, but TET was achieving the latest SoTA performance prior to LT-SNN, so if any researcher wants to advance the SoTA, we do not think there is a problem on fairness if the researcher starts from one of the SoTA techniques and builds upon it with newly proposed schemes to further improve the accuracy.
> > To that end, with new learnable threshold schemes built on top of TET, we demonstrated the outperformed accuracy of LT-SNN compared to TET, which promotes the proposed LT-SNN as the new SoTA on various event-based computer vision datasets, also outperforming DSpike [RJ5]. Unlike the differentiable forward function in DSpike [RJ5], LT-SNN directly generates the binary spike during the forward pass, which is simpler and more hardware compatible. Given the orthogonal novelties between DSpike [RJ5] and the proposed LT-SNN, we believe accuracy is the best metric of comparison.
> >
> > In the next version of our paper, we will add the citation of Dspike [RJ5] and the accuracy comparison results from the above table and description.

---

> > > ### Author Response · Authors · 2022-12-10
> > > **Response to John Willian part 3**
> > >
> > > ```
> > > **Unfair 3.** As mentioned above, TET is adopted in the proposed method and the TET baseline but is not adopted in other baseline methods. It is unfair that the authors use an advanced trick TET to beat other works that don't.
> > > ```
> > >
> > > We respectfully disagree with the remark on "It is unfair that the authors use an advanced trick TET to beat other works that don't".
> > > If any researcher wants to advance the SoTA, we do not think there is a problem on fairness if the researcher starts from one of the SoTA techniques and builds upon it with newly proposed schemes to further improve the accuracy. As TET is the current SoTA, we adopt TET loss function and incorporate it with the proposed LT-SNN. LT-SNN is novel from TET in two ways 1) it is based on the layer-wise learnable potential threshold for enhancing spike activity instead of the fixed threshold used by TET. We have endorsed the significance of the learnable threshold from the literature as mentioned in [R4, R8]. 2) We use 'SGP' for layer-wise $V_{th}$ optimization while TET is using conventional 'STSG' with a fixed threshold for all the layers that we have challenged and proved inferior with our SoTA results. Our argument is that each work adopts the existing best implementation and resolves its bottlenecks to improve SNN performance as does the LT-SNN adopting TET loss function, so it is not unfair. In the original manuscript and the previous responses, we reported the outperformed accuracy of the LT-SNN algorithm compared to the vanilla TET based on various model architectures and datasets, including small-scale DVS-CIFAR10, NCARS dataset, medium-scale NCalTech101 dataset, and also the large-scale Gen1 dataset, which proves the effectiveness and superiority of the proposed LT-SNN algorithm with practical event-based datasets.
> > >
> > > ```
> > > **Unfair 4:** I also noticed the model architecture used in Table 4 is different from prior works. It makes sense that the authors want to show the proposed method can help smaller models do better on CIFAR10DVS as claimed in Figure 1. However, to my best knowledge and experience, 1) CIFAR10DVS is a small dataset, and the obstacle is over-fitting. So, that smaller models can do better than bigger models is not surprising, and thus does not back up the proposed method effectively. 2) I think it is very necessary to use the same model architecture and hyper-parameters in the main experiments (Table 4). Just keep any other hyper-parameters consistent with the prior works. It should be common sense to make fair comparisons when proving the proposed method is better than others. The authors have done so many experiments in Table 4. Why did they lose this vital one?
> > > ```
> > >
> > > Thank you for raising this concern, Yes we agree with the reader. One of the reviewers has also raised this concern and we have conducted more experiments using the same model VGG-11 reported by DSR and TET to make an apple-to-apple comparison of LT-SNN with existing SoTAs. The table below summarizes the comparison of LT-SNN with TET and DSR for the DVS-CIFAR-10 dataset, using the identical VGG-11 model. As can be seen, LT-SNN improves the accuracy by 3.81% and 2.18% in comparison to the DSR and TET works, respectively.
> > >
> > > *[Table: Baseline comparison of proposed SGP with current SoTA.]*
> > >
> > > | **Method** | **Architecture** | **Threshold** | **SG Function** | **Top-1 Accuracy** |
> > > |:----------:|:----------------:|:-------------:|:---------------:|:------------------:|
> > > | TET[RJ1]   | VGG-11           | Fixed         | Triangle        | 77.33              |
> > > | DSR[RJ4]   | VGG-11           | Fixed         | STSG            | 75.70              |
> > > |**This work** |**VGG-11**      |**Learnable**  |**SGP**          |**79.51**           |

---

> > > > ### Author Response · Authors · 2022-12-10
> > > > **Response to John Willian part 4**
> > > >
> > > > ```
> > > > **Fairness Conclusion.** I also noticed the model architecture used in Table 4 is different from prior works. It makes sense that the authors want to show the proposed method can help smaller models do better on CIFAR10DVS as claimed in Figure 1. However, to my best knowledge and experience, 1) CIFAR10DVS is a small dataset, and the obstacle is over-fitting. So, that smaller models can do better than bigger models is not surprising, and thus does not back up the proposed method effectively. 2) I think it is very necessary to use the same model architecture and hyper-parameters in the main experiments (Table 4). Just keep any other hyper-parameters consistent with the prior works. It should be common sense to make fair comparisons when proving the proposed method is better than others. The authors have done so many experiments in Table 4. Why did they lose this vital one?
> > > > ```
> > > >
> > > > We believe that spiking neural networks will be more suitable for event-based sensors (e.g. dynamic vision sensors) and corresponding neuromorphic datasets, because such sensors output binary spikes that represent events, and these could be directly connected to the inputs of the SNN models.
> > > > On the other hand, for static image based datasets such as CIFAR-10 or ImageNet, the static images need to be converted into spike format, which incurs latency/power overhead, and this could diminish the benefits of SNNs (such overhead does not exist for conventional deep neural networks).
> > > > The spatial-temporal information captured by DVS sensors naturally fits the computation procedure of spiking neural networks, which is the major focus of the proposed LT-SNN algorithm. Also, please note that we validated the proposed LT-SNN algorithm on the large-sized Prophesee Gen1 dataset, which contains 228,123 bounding boxes for cars and 27,658 for pedestrians in the practical urban environment. The SoTA accuracy achieved by the proposed LT-SNN algorithm demonstrates the powerful performance on the large and practical event-based dataset.
> > > > Still, based on the reviewer's feedback, in the responses above, we reported the CIFAR-10 accuracy results using the identical setup with the open-sourced codes of TET [RJ1], and our LT-SNN shows superior performance in the apple-to-apple comparison.
> > > > Compared to Dspike [RJ5], LT-SNN's accuracy for DVS-CIFAR10 is 4.67% higher, and LT-SNN's accuracy for CIFAR-10 is 0.28% lower. We have provided the experimental results in the two tables of "Performance comparison of TET and LT-SNN on CIFAR-10 dataset" and "Performance comparison of TET and LT-SNN on DVS-CIFAR-10 dataset", which were presented in our responses above to the earlier comments. We will include these results and discussions in the next version of our paper.

---

> > > > > ### Author Response · Authors · 2022-12-10
> > > > > **Response to John Willian part 5**
> > > > >
> > > > > ```
> > > > > **Doubt 1.** Furthermore, DSR is proposed to enable direct-trained SNNs to have more simulation steps and meanwhile less training overhead (which is equal to only one simulation step in the backward pass). Could the proposed method achieve comparable training efficiency? If not, I think treating DSR as the main baseline could be somewhat unfair.
> > > > > ```
> > > > >
> > > > > We respectfully disagree with this comment regarding unfairness, based on the following arguments:
> > > > >
> > > > > 1) In our LT-SNN work, the main objective is to improve the performance of the post-training inference accuracy with energy-efficient model architectures.
> > > > > Even if training could be done efficiently, if the inference accuracy becomes lower due to that, the efficiency in the training process might not be able to be justified (e.g. object detection accuracy in self-driving cars).
> > > > >
> > > > > 2) From the inference perspective, we argue that the threshold-aware spiking procedure of DSR is inefficient due to the time-step-wise high-precision multiplication. Since both DSR and the proposed LT-SNN algorithm introduce the learnable membrane potential threshold into training, we believe it is fair to compare the inference accuracy and model size among different SNN algorithms. It would be unfair to alter/adjust the inference accuracy comparison based on what happens in the training process.
> > > > >
> > > > > 3) The training algorithm proposed in DSR failed to surpass the performance of the LT-SNN, where DSR exhibits lower accuracy and larger model sizes (MB) than LT-SNN, as shown in Figure 1 of our original manuscript. Furthermore, the DSR paper did not quantify the computation reduction or efficiency of the DSR algorithm, which makes it difficult to make the apple-to-apple comparison against the proposed LT-SNN algorithm.
> > > > >
> > > > > 4) With the same simulation length (T=6), the proposed LT-SNN algorithm surpasses the DSR algorithm on the CIFAR-10 dataset with the similar-size ResNet model, as shown in the table below:
> > > > >
> > > > > *[Table: Performance comparison SNN with limited simulation length on CIFAR-10.]*
> > > > >
> > > > > | **Method** | **Model** | **Simulation Length** | **Accuracy** | **Parameters** |
> > > > > |------------|-----------|-----------------------|--------------|----------------|
> > > > > | Hybrid[R8] | ResNet-18 | 10                    | 92.54        | 12.3M          |
> > > > > | tdBN[R5]   | ResNet-18 | 6                     | 93.16        | 11.17M         |
> > > > > | DSR[R6]    | ResNet-18 | 6                     | 91.89        | 11.17M         |
> > > > > | DSpike[R7] | ResNet-19 | 6                     | 94.25        | 11.18M         |
> > > > > | This work  | ResNet-19 | 6                     | 93.97        | 11.18M         |

---

> > > > > > ### Author Response · Authors · 2022-12-10
> > > > > > **Response to John Willian part 6**
> > > > > >
> > > > > > ```
> > > > > > **Doubt 2.**  Figure 2(a) seems not to match what I have observed in my experiments which show STSG performs well on the learnable potential threshold. I tried to reproduce Figure 2(a), but never find the Failed Training. It seems like Observation 1 is not generous in most scenarios. The authors' falsification of Hypothesis 1 is not persuasive since Figure 2(a) can hardly be observed. Thus, the motivation of the paper is not as concrete and solid as it claims. Will they open-source their code to reproduce Figure 2(a) or the training logs? and when? I think this is very important. Because it will be less meaningful if the whole work is based on minor special cases rather than generous circumstances.
> > > > > > ```
> > > > > >
> > > > > > Thank you for bringing this up. We had a number of experiments with STSG where we encountered the abnormality in the learnable threshold and validation accuracy as reported in Figure 2(a).
> > > > > > Based on the reviewer's comment, we shared the log files of the unsuccessful and unstable training results of STSG (of Figure 2(a)) in an anonymous Github repository: https://anonymous.4open.science/r/STSG-6B6E with VGG-7 and VGG-9 architectures on DVS-CIFAR10 dataset.
> > > > > > We will share the codes in the next version of our manuscript. On top of that, following the reviewer "bjtV" suggestion of different learning rates for weights $W$ and $V_{th}$, we performed more experiments and justified that using different learning rates could resolve the issue of abnormal training, but still at the cost of accuracy degradation as reported in the table below.
> > > > > >
> > > > > > Meanwhile, could you also share your implementation details of STSG?
> > > > > > For example, what kind of model architecture are you using, what are the hyperparameter settings of your experiments, and what is the performance of your trained model?
> > > > > > By cross-checking such training specifications and results from both sides, hopefully the discrepancy could be resolved.
> > > > > >
> > > > > > *[Table: Performance comparison of SGP and lower learning rate to balance out $V_{th}$ and $W_{ij}$ for LT-SNN training.]*
> > > > > >
> > > > > > | **Model** | **Lr-W** | **Lr-Vth** | **SGP** | **Top-1 Accuracy** | **Converged** |
> > > > > > |:---------:|:--------:|:----------:|:-------:|:------------------:|:-------------:|
> > > > > > | VGG-9     | 0.1      | 0.001      | No      | 20.40              | No            |
> > > > > > | VGG-9     | 0.001    | 0.00001    | No      | 77.20              | Yes           |
> > > > > > | VGG-9     | 0.001    | 0.001      | Yes     | 80.07              | Yes           |
> > > > > >
> > > > > > As we also responded to reviewer DoB4, we agree that Hypothesis 1 and the way of falsifying it with Figure 2(a) as an example, were not clear. To that end, to enhance clarity and avoid confusion, regarding 'Hypothesis 1' and 'Observation 1', in the next version of our manuscript, we will remove the hypothesis and observation, and instead describe the observed facts based purely on our original and newly added experimental results.
> > > > > >
> > > > > > Based on the feedback, we will open-source all the training logs for the reported results along with the codes after the decision on our submission. In addition, we will include the newly-added results in the next version of our manuscript.
> > > > > >
> > > > > > **References**
> > > > > >
> > > > > > [RJ1] S. Deng et al. "Temporal Efficient Training of Spiking Neural Network via Gradient Re-weighting." International Conference on Learning Representation (ICLR), 2022.
> > > > > >
> > > > > > [RJ2] H. Zheng et al. "Going deeper with directly-trained larger spiking neural networks". AAAI Conference on Artificial Intelligence, 2021.
> > > > > >
> > > > > > [RJ3] W. Fang et al. "Deep residual learning in spiking neural networks." In Advances in Neural Information Processing Systems (NeurIPS), 2021.
> > > > > >
> > > > > > [RJ4] Q. Meng et al. "Training High-Performance Low-Latency Spiking Neural Networks by Differentiation on Spike Representation." IEEE/CVF Conference on Computer Vision and Pattern Recognition (CVPR), 2022.
> > > > > >
> > > > > > [RJ5] Y. Li et al. "Differentiable spike: Rethinking gradient-descent for training spiking neural networks." Advances in Neural Information Processing Systems (NeurIPS), 2021.
> > > > > >
> > > > > > [RJ6] N. Rathi and K. Roy. "Diet-snn: Direct input encoding with leakage and threshold optimization in deep spiking neural networks." International Conference on Learning Representation (ICLR), 2021.
> > > > > >
> > > > > > [RJ7] G. Shen, "Backpropagation with biologically plausible spatiotemporal adjustment for training deep spiking neural networks." Patterns, 2022.

---

> ### Public Comment · ~John_Willian1 · 2023-02-06
> **Thanks for the authors' time and effort. My concerns are partially addressed. Here is my reply for the authors' response and request.**
>
> Thanks for the authors' time very much. I appreciate it! But, they did not respond to me before the public discussion period ended. Consequently, I could not respond to the authors' rebuttal in time.
>
>
>
> Nevertheless, I want to give my response now, despite the conference's decision for this paper being made.  Because I want to be a reminder for any other researcher who possibly queries this web page and to be of some help to this work.
>
>
>
> Some of my concerns are solved by the authors' rebuttal, but others are not.
>
>
>
> (Possibly Convincing 1)
>
> The authors' response to my **Unfair 1** is personally accepted. The reproducibility of TET is doubtable to me. To verify the authors' rebuttal, I also tried reproducing TET with the official code on CIFAR10. The result is even worse than what is reported in the authors' comments. I guess the official code of TET might have hidden some tricks or hyper-parameters.  I did not pay much attention to tuning the TET-related parameters.  But, I noticed the existence of successful reproduction [here](https://openreview.net/forum?id=_XNtisL32jv).
>
>
>
>
>
> (Convincing 2)
>
>
>
> The authors' response to my **Unfair 4** is accepted. **79.51** should be the SoTA for now. Please make sure it is reproducible.
>
>
>
>
>
> (Convincing 3)
>
>
>
> The authors' response to my **Doubt 1** is accepted. I read the content related to DSR again. The authors are trying to prove that the learning strategy of DSR for threshold is invalid. It makes sense that DSR fails to utilize the learnable threshold and motivates the authors to find out a proper way. And then, the authors tried STSG, which didn't work well.
>
>
>
> Such a start to the paper is reasonable and idiomatic.
>
>
>
> (Unconvincing 1)
>
> The authors' response to my **Unfair 2 & 3** is unacceptable.  *[*Table: Performance comparison of TET and LT-SNN on DVS-CIFAR-10 dataset.*]* still use different model architectures. The authors seem to misunderstand my point and stick to the wrong mindset.
>
>
>
> To **achieve SoTA performance**, adding many advanced tricks is certainly **OK**.  But, when it comes to **comparing** with others, the essential motivation is to prove that your method is more **effective**. **What if Dspike + TET would be better than LT + TET?**
>
>
>
> When proving that objective B is more effective than A, just keeping objective A substituted with B while other uninvolved hyper-parameters are unchanged. For any **fair** comparisons in any study, it should be common sense. I am actually shocked by the authors' comments that they think it is fair to `` use an advanced trick TET to beat other works that don't.``
>
>
>
> I think it would be convincing if the main experiments were based on the LT-SNN alone to demonstrate the effectiveness of LT-SNN.  Based on this foundation, the authors would be appreciated to further demonstrating that LT-SNN is more advanced because LT + TET could achieve SoTA results on various benchmark datasets.
>
>
>
> (Discussion 1)
>
> The authors' response to my **Doubt 2** is acceptable but not satisfactory. Removing the hypothesis and observation of the **Failed Training** is correct. Keeping them in another format in the main content may be inappropriate.
>
>
>
> I didn't shrink any initial learning rates as the authors did in the rebuttal, nor find any **Failed Training** either.
>
>
>
> The training logs and my implemented LT-neuron can be found [here](https://drive.google.com/drive/folders/1tyk6cBTN8LDp_8pATiXUVMlguZQhmjU_?usp=sharing).  Except for different GPU servers, I keep the hyper-parameters and the experimental settings (e.g. scheduler, optimizer, model parameters, etc.) consistent with the open-source SNN implementation [here](https://github.com/Ikarosy/Gated-LIF). My training scripts are based on SpikingJelly, which is frequently used and highly recommended in recent SNN research.
>
>
>
>  I can't open-release my whole scripts because the codes are used in an in-progress project that I wouldn't like to share. But, I believe any verifier can easily try any LT-based model or dataset by following my reproduction method as described above: [Hyperparameters & Experimental Settings](https://github.com/Ikarosy/Gated-LIF) + [SpikingJelly](https://github.com/fangwei123456/Parametric-Leaky-Integrate-and-Fire-Spiking-Neuron) + STSG-based LT-neuron ([mine](https://drive.google.com/drive/folders/1tyk6cBTN8LDp_8pATiXUVMlguZQhmjU_?usp=sharing) or rewritten by your own). I believe the outcome should be the same as mine. I recommend the authors fine-tune their experimental settings, such as LR scheduler, optimizer, etc, to check out whether the **Failed Training** is a general phenomenon.
>
>
>
> In my opinion, the so-call **Failed Training** is a minor special case due to **improper experimental settings**.  It is inappropriate to build the whole work on this minor case. I think the essential motivation of this work should be "how to leverage the adaptive threshold to achieve better performance".

---

### Comment · Area_Chair_j6XY · 2022-12-10
**The authors posted new rebuttals**

Dear Reviewers,
The authors have posted new rebuttals. I hope you could have a look and give your final opinion. Thanks!

AC

---

### Decision · Program_Chairs · 2023-01-20

**Decision:**

Reject

**Justification For Why Not Higher Score:**

The paper should be rejected for sure.

**Justification For Why Not Lower Score:**

N/A

**Metareview: Summary, Strengths And Weaknesses:**

The paper originally got two 8s (accept, good paper), one 5 (marginally below threshold) and two 3s (reject). The major challenges inclide unconvincing (and even unfiar) experimental comparisons, unconvincing motivation of the proposed method, unsatisfactory paper writing, limited novelty, etc. There are many discussion between the authors and reviewers. After rebuttal, one reviewer agreed on the novelty and experiment issues and downgraded from 8 to 5, while the other reviewer raised from 3 to 5. Since only one reviewer was positive towards the paper, the AC deemed that the paper is not good enough for ICLR, thus recommended rejection, yet thanked the authors for taking great efforts on the rebuttal.

**Summary Of Ac-Reviewer Meeting:**

N/A